# Structural and mechanistic insights into *Streptococcus pneumoniae* NADPH oxidase

**Victor R. A. Dubach** [1,2,4], **Pablo San Segundo-Acosta** [1,3,4] ✉ & **Bonnie J. Murphy** [1] ✉

Nicotinamide adenine dinucleotide phosphate (NADPH) oxidases (NOXs) have a major role in the physiology of eukaryotic cells by mediating reactive oxygen species production. Evolutionarily distant proteins with the NOX catalytic core have been found in bacteria, including *Streptococcus pneumoniae* NOX (SpNOX), which is proposed as a model for studying NOXs because of its high activity and stability in detergent micelles. We present here cryo-electron microscopy structures of substrate-free and nicotinamide adenine dinucleotide (NADH)-bound SpNOX and of NADPH-bound wild-type and F397A SpNOX under turnover conditions. These high-resolution structures provide insights into the electron-transfer pathway and reveal a hydride-transfer mechanism regulated by the displacement of F397. We conducted structure-guided mutagenesis and biochemical analyses that explain the absence of substrate specificity toward NADPH and suggest the mechanism behind constitutive activity. Our study presents the structural basis underlying SpNOX enzymatic activity and sheds light on its potential in vivo function.

Reactive oxygen species (ROS), such as superoxide anion radical ($O_2^{\cdot-}$) and hydrogen peroxide ($H_2O_2$), are highly reactive chemical species that have essential roles in immunity, cell signaling, aging and cancer[1]. The nicotinamide adenine dinucleotide phosphate (NADPH) oxidase (NOX) family of enzymes (enzyme class 1.6.3.1) generate ROS as their primary product and, therefore, have a major role in ROS homeostasis. NOXs belong to the ferric reductase transmembrane component-like domain (FRD) superfamily and are composed of a catalytic core with two conserved domains: a domain with six transmembrane helices (TM domain) coordinating two hemes by four conserved histidine residues and a C-terminal cytosolic dehydrogenase domain (DH domain) containing the flavin adenine dinucleotide (FAD)-binding and NADPH-binding sites (Fig. 1a). The DH domain catalyzes a hydride transfer to FAD, which subsequently transfers electrons stepwise to the two transmembrane hemes. The outer heme reduces oxygen to generate $O_2^{\cdot-}$ or $H_2O_2$ (refs. 2,3).

Although NOXs were first thought to be exclusive to eukaryotic genomes, genes encoding proteins sharing the same catalytic core have more recently been found in bacteria[4,5]. With the exception of cyanobacterial NOX5, which seems to have arisen through horizontal gene transfer from eukaryotic donors, bacterial NOX homologs are evolutionarily distant from eukaryotic NOXs and consist of a single polypeptide chain with DH and TM domains[5]. The best-characterized member is the *Streptococcus pneumoniae* NOX (SpNOX). SpNOX is a 46-kDa protein that can be produced with correct heme incorporation in high yields and shows high stability and robust activity when solubilized in lauryl maltose neopentyl glycol (LMNG)[4,6].

Structural studies of NOX proteins are generally difficult because of low yields after recombinant expression, loss of cofactors during purification and flexibility. The first structure of a NOX protein was the crystal structure of separate TM and DH domains from *Cylindrospermum stagnale* NOX5 (csNOX5), which gave insight into the catalytic core and the oxygen-reduction center[7]. However, despite notable efforts, obtaining full-length NOX crystals diffracting to high resolution has proven to be challenging[6,7]. Nevertheless, the use of cryo-electron microscopy (cryo-EM) has allowed the structures of

[1]Redox and Metalloprotein Research Group, Max Planck Institute of Biophysics, Frankfurt am Main, Germany. [2]Redox and Metalloprotein Research Group, IMPRS on Cellular Biophysics, Frankfurt am Main, Germany. [3]Present address: Chronic Disease Programme, UFIEC, Carlos III Health Institute, Madrid, Spain. [4]These authors contributed equally: Victor R. A. Dubach, Pablo San Segundo-Acosta. ✉e-mail: psegundo@isciii.es; bonnie.murphy@biophys.mpg.de

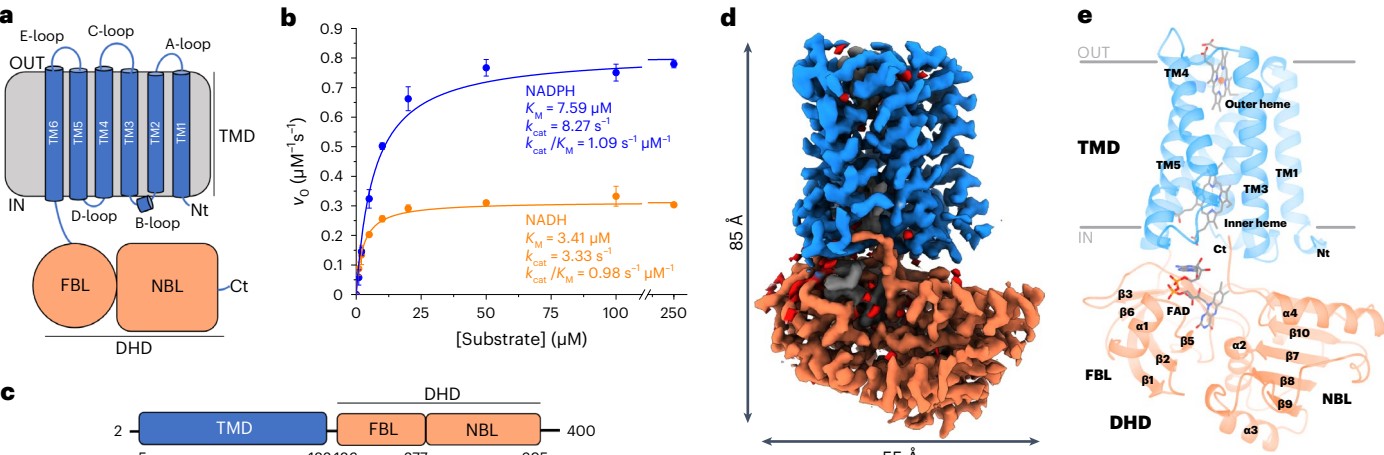

**Fig. 1 | SpNOX displays the canonical NOX structure and NOX activity. a**, SpNOX is composed of a TM domain (TMD) and a DH domain (DHD) subdivided into an FAD-binding lobe (FBL) and an NADPH-binding lobe (NBL). Nt, N terminus; Ct, C terminus. **b**, The activity of SpNOX under steady-state conditions was measured using a cytochrome c reduction assay. Apparent $K_M$ and $k_{cat}$ values were obtained by fitting the data to the Michaelis–Menten equation. The mean values of three technical replicates are plotted and the s.d. is indicated. **c**, Schematic view of SpNOX domain boundaries. **d**, Side view of the cryo-EM map of substrate-free SpNOX. The TMD and DHD are colored blue and coral, respectively. The same coloring is used throughout the article unless otherwise stated. **e**, Structure of substrate-free SpNOX in cartoon representation. Membrane boundaries are represented as gray lines.

mouse and human dual oxidases (DUOX1)[8,9] and the inactive human core NOX2 complex[10,11] to be solved. These structures have provided essential information about the subunit interactions and catalytic regulation of eukaryotic NOXs but large questions remain regarding the mechanism of activation and catalytic turnover. Moreover, no structure of a NOX-like bacterial protein is currently available. Because of the relatively simple expression and purification of SpNOX and its homology to eukaryotic NOX proteins, a structural characterization of SpNOX would be valuable for its application as a model system of NOXs.

Here, we present high-resolution cryo-EM structures of SpNOX obtained under substrate-free conditions, under stably reducing conditions with nicotinamide adenine dinucleotide (NADH) and under turnover conditions with NADPH. Compared to csNOX5, DUOX1 and NOX2, SpNOX is smaller, has shorter extracellular loops and, like the mammalian NOX4, is constitutively active. These structures characterize a bacterial NOX-like protein and, together with structure-based mutagenesis, provide insights into the electron-transfer pathway, catalytic mechanism and constitutive activity.

## Results

### SpNOX purification, analysis and structure determination

*S. pneumoniae* NOX was expressed in *Escherichia coli* and purified as previously described with minor modifications (Methods)[4,6]. The protein was solubilized using LMNG and showed a single band in SDS–PAGE and a homogeneous single peak on size-exclusion chromatography, corresponding to the monomeric protein (Supplementary Fig. 1a,b). Quick removal of imidazole after Ni-NTA purification was crucial to avoid protein aggregation. The ultraviolet–visible light (UV–vis) spectrum of the purified protein showed the characteristic Soret peak at 414 nm, indicating correct heme incorporation (Supplementary Fig. 1c). Oxidation of NADPH could be directly followed by monitoring absorbance at 340 nm under aerobic conditions. A similar experiment in the absence of $O_2$ showed only initial activity because of residual $O_2$ in the buffer and/or reduction of SpNOX cofactors before the oxidation activity ceased, confirming that $O_2$ can act as an electron acceptor for SpNOX (Supplementary Fig. 1d). NADPH oxidation was also measured with a cytochrome c reduction assay[4] (Fig. 1b) and Michaelis–Menten analysis gave an apparent $K_M$ value of 7.59 μM and an apparent $k_{cat}$ of 8.27 s$^{-1}$. The protein also exhibited NADH-driven

activity, with an apparent $K_M$ value of 3.41 μM and an apparent $k_{cat}$ of 3.33 s$^{-1}$. Cytochrome c reduction assays under anaerobic conditions revealed direct electron transfer to cytochrome c (Supplementary Fig. 1e), likely from the outer heme, and to superoxide dismutase (SOD) (Supplementary Fig. 1f). Therefore, the rate of cytochrome c reduction by the anion superoxide could not be estimated with SOD inhibition. Our results, obtained under both aerobic and anaerobic conditions, indicate that previous efforts to measure specific superoxide production using SOD inhibition may have been subject to unforeseen artifacts. The initial rates of cytochrome c reduction under aerobic and anaerobic conditions were very similar (Supplementary Fig. 1g), suggesting cytochrome c acts as the major direct electron acceptor in the in vitro assay. Contrary to previous observations[4], we could observe ferric reductase activity under anaerobic conditions, indicating direct electron transfer to ferrous iron (Supplementary Fig. 1h). However, the direct transfer rate of SpNOX to ferrous iron (3.6 min$^{-1}$) was much lower than to cytochrome c (124.7 min$^{-1}$) or molecular oxygen (48 min$^{-1}$) at similar initial concentrations. Moreover, Michaelis–Menten analysis of ferric reductase activity showed apparent $k_{cat}$ (0.05 s$^{-1}$) and $K_M$ (81.3 μM) values corresponding to a catalytic efficiency of 0.0006 s$^{-1}$ μM$^{-1}$ (Supplementary Fig. 1i). These experiments suggest that the ferric reductase activity of SpNOX might not be physiologically relevant but rather an effect of the promiscuity of SpNOX for the final electron acceptor in vitro.

Cryo-EM single-particle analysis of SpNOX was performed in LMNG micelles and showed that the protein preparation was homogeneous (Methods and Supplementary Figs. 2–5). In addition to a structure of SpNOX in the absence of electron donor, we also determined structures of the stably reduced protein bound to NADH under anaerobic conditions and of the wild-type (WT) protein and an F397A mutant under turnover with NADPH and $O_2$ present. Despite the relatively small size of the protein (46 kDa) for single-particle analysis, the final maps reached nominal resolutions ranging from 2.2 to 2.6 Å (Table 1 and Supplementary Figs. 6 and 7), allowing us to resolve essentially all residues. The TM and DH domains reached similar overall local resolutions, with only small regions of the NADPH-binding lobe being less well resolved (Supplementary Fig. 6). In contrast to previously published small-angle neutron scattering data indicating potential interdomain flexibility[6], our results indicate that the DH domain interacts rigidly with the TM domain, thus facilitating particle alignment.

**Table 1 | Cryo-EM data collection, refinement and validation statistics**

| | Substrate-free (EMD-18644), (PDB 8QT6) | NADPH-bound (EMD-18645), (PDB 8QT7) | NADH-bound (EMD-18646), (PDB 8QT9) | F397A(EMD-18647), (PDB 8QTA) |
|---|---|---|---|---|
| **Data collection and processing** | | | | |
| Magnification | ×215,000 | ×215,000 | ×215,000 | ×215,000 |
| Voltage (kV) | 300 | 300 | 300 | 300 |
| Electron exposure (e⁻ per Å²) | 70 | 70 | 70 | 70 |
| Defocus range (µm) | −0.7 to −2.1 | −0.7 to −2.1 | −0.7 to −2.1 | −0.7 to −2.1 |
| Pixel size (Å) | 0.573 | 0.573 | 0.573 | 0.573 |
| Symmetry imposed | C1 | C1 | C1 | C1 |
| Initial particle images (no.) | 3,376,347 (Topaz) 3,115,826 (Topaz) 2,273,739 (crYOLO) | 6,641,288 (Topaz) 4,948,429 (Topaz) 2,868,991 (crYOLO) | 5,571,895 (Topaz) 5,183,375 (Topaz) 3,583,885 (crYOLO) | 5,003,252 (Topaz) 4,668,309 (Topaz) 4,145,972 (crYOLO) |
| Final particle images (no.) | 397,972 | 591,137 | 697,211 | 546,234 |
| Map resolution (Å) | 2.29 | 2.20 | 2.36 | 2.64 |
| FSC threshold | 0.143 | 0.143 | 0.143 | 0.143 |
| Map resolution range (Å) | 2.25–26.65 | 2.05–32.20 | 2.11–36.46 | 2.50–9.30 |
| **Refinement** | | | | |
| Initial model used | AF model (AF-Q8CZ28) | Substrate-free model | Substrate-free model | NADPH-bound model |
| Model resolution (Å) | 2.44 | 2.33 | 2.44 | 2.75 |
| FSC threshold | 0.5 | 0.5 | 0.5 | 0.5 |
| Map sharpening *B* factor (Å²) | −80.1 | −74.8 | −79.9 | −105.4 |
| **Model composition** | | | | |
| Nonhydrogen atoms | 3,466 | 3,516 | 3,511 | 3,472 |
| Protein residues | 399 | 399 | 399 | 399 |
| Ligands | | | | |
| HEM | 2 | 2 | 2 | 2 |
| FAD | 1 | 1 | 1 | 1 |
| NDP | – | 1 | – | 1 |
| NAI | – | – | 1 | – |
| Waters | 63 | 65 | 64 | 27 |
| *B* factors (Å²) | | | | |
| Protein | 34.55 | 41.92 | 37.17 | 49.30 |
| Ligand | 22.74 | 34.50 | 28.78 | 46.19 |
| Water | 31.09 | 40.97 | 35.05 | 40.24 |
| Root-mean-square deviations | | | | |
| Bond lengths (Å) | 0.004 | 0.009 | 0.003 | 0.009 |
| Bond angles (°) | 0.940 | 0.931 | 0.578 | 0.940 |
| Validation | | | | |
| MolProbity score | 1.76 | 1.27 | 1.26 | 1.56 |
| Clashscore | 4.72 | 4.96 | 5.55 | 7.45 |
| Poor rotamers (%) | 0.86 | 0.86 | 0.58 | 1.16 |
| Ramachandran plot | | | | |
| Favored (%) | 98.24 | 98.24 | 98.74 | 97.48 |
| Allowed (%) | 1.76 | 1.76 | 1.26 | 2.52 |
| Disallowed (%) | 0 | 0 | 0 | 0 |
| Q-score | 0.82 | 0.83 | 0.82 | 0.77 |

PDB, Protein Data Bank; FSC, Fourier shell correlation; HEM, heme group; NDP, NADPH; NAI, NADH.

While this paper was under review for publication, a study describing high-resolution crystal structures of the substrate-free WT and F397W DH domains and a low-resolution F397W full-length inactive structure of SpNOX became available as a preprint[12].

### The architecture of SpNOX
The structure of SpNOX corresponds to the canonical NOX structure, with an N-terminal TM domain that coordinates two hemes and a C-terminal DH domain bound to FAD (Fig. 1c–e). Despite sharing an

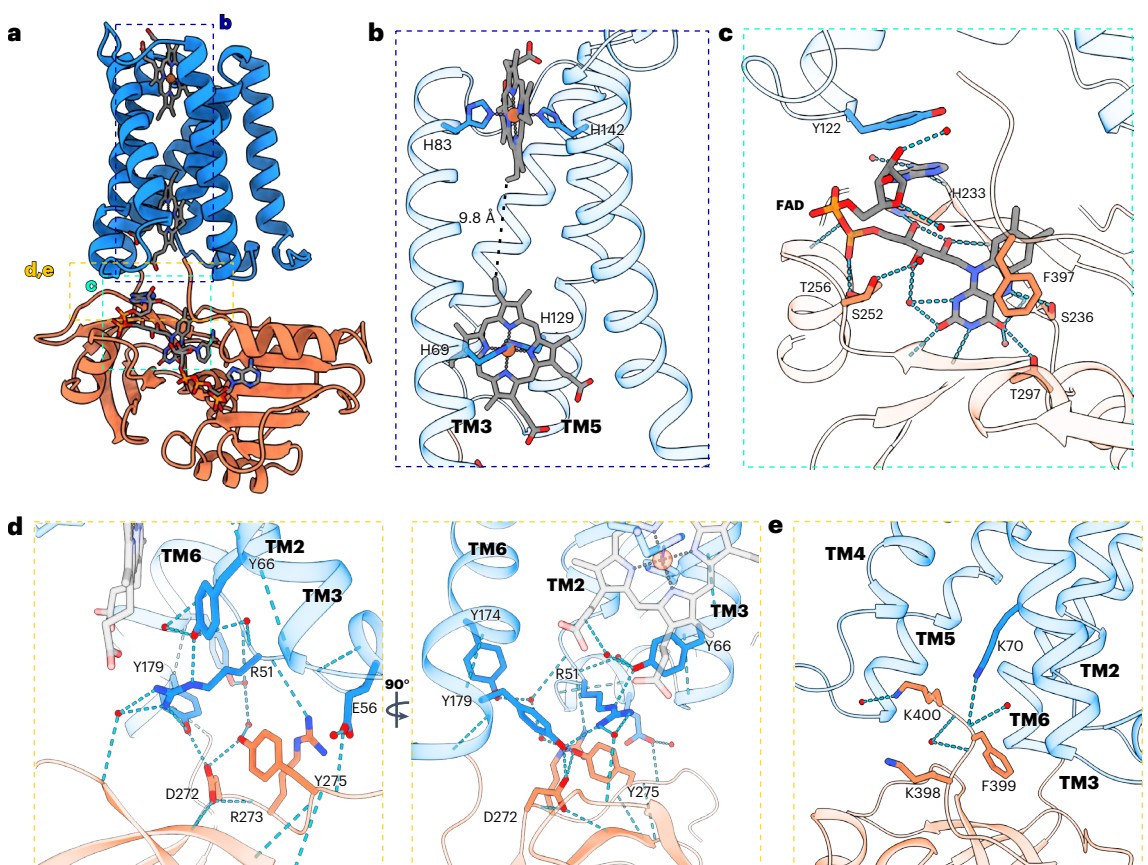

**Fig. 2 | Heme-binding and FAD-binding sites and interactions between the TM and DH domains. a**, Cartoon representation of the structure of NADPH-bound SpNOX. Areas of interest are highlighted with colored dashed boxes. **b**, The two B-type hemes are coordinated by two histidine pairs H83 + H142 (outer heme) and H69 + H129 (inner heme) of the transmembrane helices TM3 and TM5. The edge-to-edge distance is indicated with a dashed line. TM1 and TM2 are omitted for clarity. **c**, Detailed view of the FAD-binding site. Side chains and FAD are shown as sticks. **d**, Closer look into the interface interactions between the TM and the DH domains. In **c**,**d**, water molecules are shown as red spheres and atoms within H-bond distance are marked with cyan dashed lines. **e**, The C-terminal tail of SpNOX lies at the interface of the TM and DH domains, is oriented orthogonal to the membrane plane and interacts by H-bonds between K70 at TM3 and F399.

overall sequence identity between 10% and 14% with human and cyanobacterial NOXs (Supplementary Fig. 8a), the SpNOX structure strongly resembles the structures of eukaryotic NOXs and csNOX5 TM and DH domains (Supplementary Fig. 9).

The SpNOX TM domain encompasses six transmembrane helices (TM1–TM6) with an overall pyramidal shape, triangular on the inner membrane side and narrower toward the extracellular space. This folding strongly resembles the ferric reductase domain of eukaryotic NOXs, with TMs 2–5 adopting an hourglass-shaped conformation that binds two B-type hemes orthogonal to the membrane plane, one located closer to the cytosolic side (inner heme) and the other closer to the outer side (outer heme). Both are hexacoordinated by two pairs of histidines (H69 and H129, inner heme; H83 and H142, outer heme) belonging to TM3 and TM5 (Fig. 2a,b and Supplementary Fig. 8b). The iron-to-iron distance is 21.47 Å, while the edge-to-edge distance is 9.84 Å. Interestingly, the outer heme of SpNOX is flipped ~180° with respect to eukaryotic NOXs (Supplementary Fig. 9) and this seems to be determined by the residues interacting with the propionate groups in the structure (Supplementary Fig. 8a).

The C-terminal DH domain is connected to TM6 by a short linker and shows the canonical ferredoxin–NADP+ reductase (FNR) fold: an FAD-binding lobe consisting of six antiparallel β-strands forming a β-barrel flanked by an α-helix and an NADPH-binding lobe with a Rossman-like topology formed by a parallel β-sheet sandwiched between α-helices (Fig. 2a and Supplementary Fig. 10a). When compared to eukaryotic NOXs and csNOX5 (Supplementary Figs. 8c and 9),

the DH domain of SpNOX shows up to 19% sequence identity with a simplified overall architecture. The main structural differences accumulate at the NADPH-binding lobe (Supplementary Fig. 9), where SpNOX shows an unstructured long linker connecting the parallel β-strands. In eukaryotic NOXs, insertions and rearrangements at this site serve important regulatory functions (for example, the calmodulin-binding region of NOX5).

SpNOX displays high affinity for FAD (~60 nM) and lower affinity for other smaller flavins such as flavin mononucleotide[12]. A clear density for FAD was well resolved in all maps (Supplementary Fig. 7) inside a positively charged pocket at the interface of the FAD-binding lobe and the TM domain (Supplementary Fig. 10b). The DH domain residues H233, S236, K250, S252, T256 and T297 interact with FAD by hydrogen bonds (H-bonds), whereas F397 interacts by π–π stacking with the isoalloxazine ring (Fig. 2c and Supplementary Fig. 10c). Residues H233 and S236 belong to the strictly conserved 'HPF(S/T)' motif (Supplementary Fig. 8c). The only direct interaction between FAD and the TM domain is through π–π stacking between the adenine and Y122 (Fig. 2c and Supplementary Fig. 10d), which is strictly conserved in bacterial NOX-like proteins (Supplementary Fig. 11). A Y122A mutant was characterized in the literature[12] and showed a reduced apparent affinity for FAD and a reduced $k_{cat}$, confirming the importance of this residue in FAD binding. The adopted geometry of FAD is the same as in NOX2, which achieves this interaction through the Y122-equivalent residue F202 (Supplementary Fig. 10e). Interestingly, this FAD conformation is also present in human DUOX1, in spite of the fact that different interactions

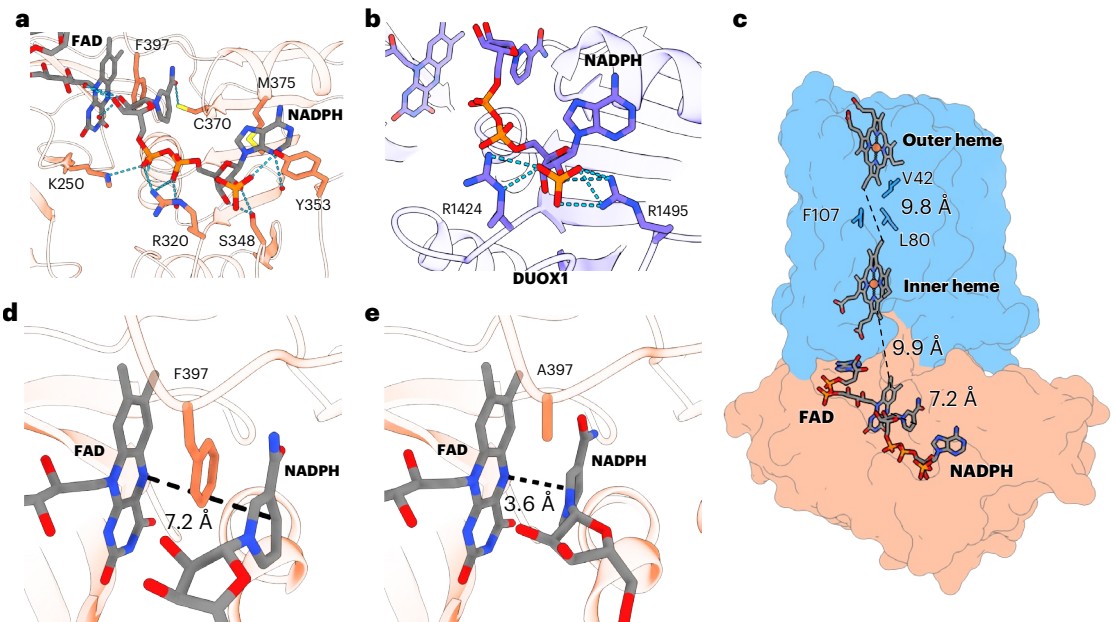

**Fig. 3 | The NADPH-binding site and the SpNOX electron-transfer pathway.**
**a**, Detailed view of the NADPH-binding site at the DH domain of SpNOX. Atoms within H-bond distance are marked with cyan dashed lines. FAD (gray) and amino acid side chains (coral) are shown as sticks. **b**, The lack of specificity toward NADPH in SpNOX can be explained by the absence of ionic interactions with the 2′-phosphate, unlike in eukaryotic NOXs including human DUOX1 (PDB 7D3F)[9], in which R1424 and R1495 interact with the 2′-phosphate. **c**, The proposed electron-transfer path within SpNOX. Hemes, FAD, NADPH and the inter-heme

hydrophobic residues are shown as sticks on the surface of SpNOX. Distances between the redox cofactors are represented as dashed black lines and were measured between the nicotinamide and the isoalloxazine ring (7.2 Å), between the isoalloxazine ring and the inner heme lower edge (9.9 Å) and between the edges of the inner and outer hemes (9.8 Å). **d**, F397 sits between the isoalloxazine ring of FAD and the nicotinamide ring of NADPH, impeding hydride transfer. **e**, The nicotinamide ring of NADPH moves closer to the isoalloxazine group of FAD in the F397A SpNOX mutant.

(mainly salt bridges linking the phosphate from adenine monophosphate (AMP) with R1214 and R1131 of the TM domain) are responsible for forming the interaction with FAD (Supplementary Fig. 10f).

The DH domain is docked through direct polar interactions of residues at the FAD-binding lobe, mainly with residues located at the B-loop and TM6 of the TM domain (Fig. 2d). Moreover, K70 at TM3 interacts by H-bonds with F399, which, together with K398 and K400, is part of the C-terminal tail of SpNOX (Fig. 2e). A Δ398–400 mutant showed an increased $K_M$ for NADPH and a reduced $k_{cat}$ (Supplementary Fig. 14a), indicating that these residues might be relevant for the docking of the TM and DH domains and for NADPH binding.

**SpNOX NADPH-binding site and electron-transfer pathway**
NADPH-bound (2.2-Å resolution) and anaerobic NADH-bound (2.4-Å resolution) SpNOX maps showed well-resolved densities for the substrate (Fig. 3 and Supplementary Fig. 7). Aside from the bound substrate, all three structures appear largely the same with only minor side chain rearrangements (Supplementary Fig. 12a–c,e–g,i–k). The nicotinamide moiety of NADPH and NADH is accommodated by H-bonds and hydrophobic interactions inside a cavity generated by F397 at the C-terminal tail and the strictly conserved 'XGXGX' and 'CG(S/P)' motifs located at the loop connecting α2 and β8 and at the loop connecting β10 and α3, respectively (Fig. 3a and Supplementary Figs. 8c, 12d,h,l and 13a). A C370A mutant showed an increase in $K_M$, confirming the relevance of the 'CG(S/P)' motif for substrate binding (Supplementary Fig. 14b).

Overall, the AMP-binding site of both substrates shows the highest number of interactions with SpNOX. Previous structural studies of NADPH-specific FNR superfamily members, such as DUOX1 (ref. 9), *Anabaena* PCC 7119 FNR[13] and human neuronal nitric oxide synthase (nNOS)[14], showed that the AMP-binding region is critical for coenzyme specificity because of the presence of at least one arginine residue that establishes charge-to-charge interactions with the 2′-phosphate group

of NADPH. In SpNOX, this group is coordinated by H-bonds with S348 and Y353 (Fig. 3a), which is remarkably different to the NADPH-binding mode of eukaryotic NOXs and other members of the NADPH-specific FNR superfamily. In human DUOX1, two arginine residues, one analogous to R320 of SpNOX (R1424) and another analogous to Y353 (R1495), both coordinate the 2′-phosphate by salt bridges (Fig. 3b). In human NOX2, R513 also occupies the position of Y353 (Supplementary Fig. 8c). In *Anabaena* FNR, a tyrosine residue analogous to Y353 interacts with the adenine moiety; however, as in eukaryotic NOXs, the 2′-phosphate is also coordinated by salt bridges with two arginine residues (Supplementary Fig. 13b). Therefore, NADPH specificity results, at least partially, from ionic interactions that are absent in SpNOX, which would explain the lack of substrate specificity. Supporting this, a Y353R mutant increased the specificity of SpNOX for NADPH fourfold (Supplementary Fig. 14c,d). Additionally, the 5′phosphate interacts by a salt bridge with R320, the adenine moiety is sandwiched between the M375 side chain and the aromatic ring of Y353 and the 3′-OH of the ribose performs H-bonding with S318 and S348. The nicotinamide-bound ribose is hydrated by several resolved water molecules but does not show any direct interaction with SpNOX (Supplementary Fig. 13c). The comparison of the low-resolution SpNOX crystal structure, high-calcium NADPH-bound DUOX1 and inactive NOX2 from previous studies[9,12] suggests that eukaryotic NOXs may require a tighter interaction between the NADPH-binding and FAD-binding lobes instead of a large domain motion for efficient hydride transfer. Our comparison of NADPH-bound SpNOX and high-calcium human DUOX1 (Supplementary Fig. 13d,e) further supports this idea and suggests a potential role of an NOX-conserved positively charged residue (DUOX R1337 and SpNOX K250; Supplementary Fig. 8c) that allows the nicotinamide ring to approach the isoalloxazine ring by forming a salt bridge with the phosphate adjacent to the ribose (Supplementary Fig. 13f).

The apparent electron-transfer pathway of SpNOX corresponds to the previously described electron pathways of NOX proteins: NADPH →

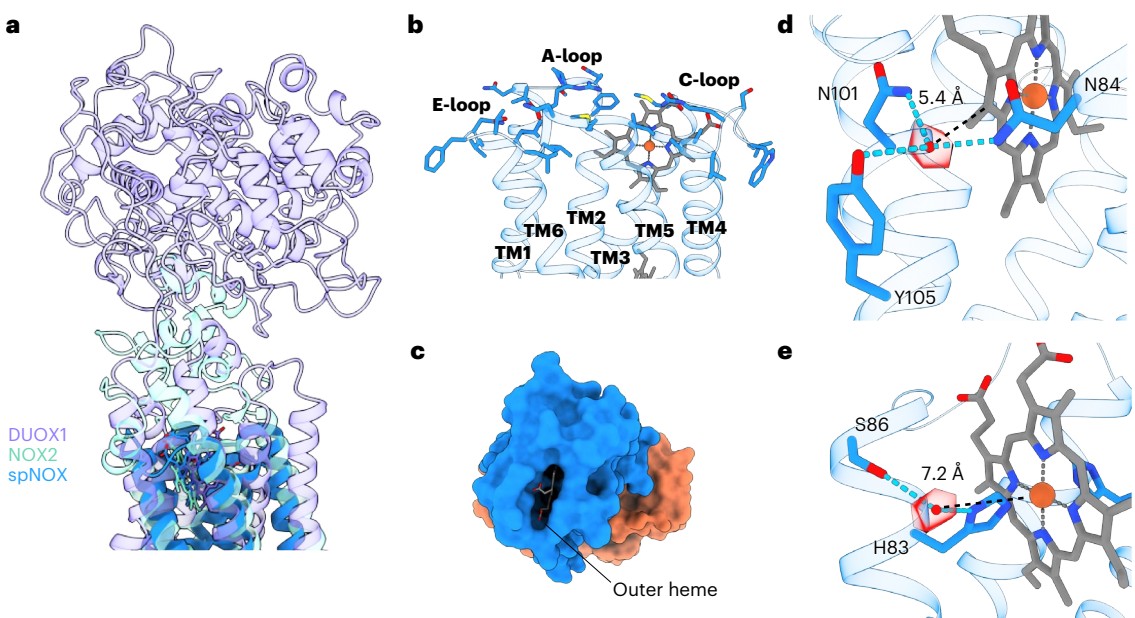

**Fig. 4 | SpNOX displays a highly solvent-exposed outer heme and two putative oxygen-binding sites. a**, SpNOX (cartoon representation, blue) presents an outer heme more solvent exposed than eukaryotic NOXs such as NOX2 (cyan; PDB 8GZ3)[10] or DUOX1 (purple; PDB 7D3F)[9] because of a lack of extracellular domains or large loops. **b**, Side view of the extracellular loops of SpNOX. **c**, Top view of SpNOX surface showing the solvent-exposed outer heme. **d,e**, Two different putative oxygen-reduction centers were identified in SpNOX, one formed by N84, N101 and Y105 (**d**), and another formed by H83 and S86 (**e**). The distance between the coordinated water molecules (red sphere and density) and the outer heme is shown as a black dashed line. Atoms within H-bond distance are marked with cyan dashed lines.

FAD → inner heme → outer heme → $O_2$ (refs. 7–11). The distance between the isoalloxazine ring and the lower edge of the inner heme is 9.9 Å, whereas the edge-to-edge distance between hemes is 9.8 Å (Fig. 3c). The space between the two hemes is partially occupied by a cluster of hydrophobic amino acids that includes the aromatic residues F107 and Y136. F107 occupies a similar position to a conserved aromatic residue present in other NOXs (F215 in human NOX2, F1097 in human DUOX1 and W378 in csNOX5). Mutational analysis in mouse DUOX1 and NOX2 (ref. 8,11) showed that this amino acid could be the preferred route for electron transfer between the hemes. Here, we analyzed the activity of an F107L mutant and an F107L;Y136L double mutant (Supplementary Fig. 14e,f). Contrary to the previous observations in eukaryotic NOXs, we did not detect any notable reduction of SpNOX activity. These results indicate that, in SpNOX, neither F107 nor Y136 is required for efficient electron transfer and suggest that the loss of activity in other NOXs could be the consequence of a less optimal hydrophobic environment or structural rearrangements. This is in agreement with previous observations that electron tunneling between redox centers separated by distances below 14 Å occur at rates fast enough not to be a limiting factor for substrate turnover[15].

Our substrate-bound structures represent an inactive state of the enzyme, as deduced by a 7.2-Å core-to-core distance between the nicotinamide C4 and the isoalloxazine N5, too long for efficient hydride transfer (Fig. 3c,d), which should require distances consistent with simultaneous bond breakage and formation in the transition state[16–18]. The fact that a productive structure could not be solved may indicate that the active state is transient and, thus, sparsely populated[19,20]. In our inactive conformations, F397 is stacked between the nicotinamide and the isoalloxazine rings (Fig. 3d), suggesting that during turnover a conformational change involving, at least, the displacement of the F397 lateral chain is required for hydride transfer. Remarkably, an analogous C-terminal aromatic residue is conserved within other members of the FNR superfamily including plant-type FNRs, NOSs and cytochrome P450 reductases[14,21–23]. Extensive structural and biochemical studies have suggested that the displacement of this residue is

highly thermodynamically unfavored and may act as the rate-limiting step for flavin reduction[21,24–26]. This residue may also contribute to the regulation of NADPH-binding affinity and specificity and to the stabilization of the FAD semiquinone state[24]. Substitution of the C-terminal tyrosine of plant-like FNRs to a nonaromatic residue such as alanine or serine substantially increased the affinity for $NADP^+$ and NADPH and induced an enzyme state with productive flavin–nicotinamide interaction[18]. However, in other FNR superfamily members such as human nNOS, such a substitution did not lead to a large change in the binding affinity for NADPH, although it increased the $k_{cat}$ for NADH[22]. Here, we analyzed the steady-state kinetics of an F397A SpNOX mutant (Supplementary Fig. 14g). Similarly to nNOS, we did not observe a large effect on the $K_M$, which suggests that F397 does not have a major role in the control of NADPH-binding affinity in SpNOX. As for the F397S SpNOX mutant described in the previous study[12], we observed a moderate increase in $k_{cat}$, which could be explained by the elimination of the F397 displacement step. These data are supported by further analysis of a cryo-EM structure of the F397A mutant bound to NADPH under turnover conditions at 2.64-Å resolution (Table 1, Fig. 3e and Supplementary Fig. 5). Density can be observed for the bound substrate but, compared to the other substrate-bound structures described here, the nicotinamide is closer to the isoalloxazine ring (Fig. 3e and Supplementary Fig. 12i–k). The absence of the large side chain of F397 allows NADPH to take up a productive conformation in SpNOX, as seen previously for pea FNR Y308 mutants[18]. In this conformation, the nicotinamide ring does not lie parallel to the isoalloxazine ring but at a -26° angle (Fig. 3e).

A search for potential oxygen-reduction centers close to the outer heme-binding pocket revealed a strikingly high degree of exposure of the outer heme to the extracellular space, which is more buried within the structure of other NOXs by long extracellular loops or domains (Fig. 4a)[8,11]. In SpNOX, the extracellular loops (A-loop, C-loop and E-loop) fold away from the heme cavity, directly exposing the outer heme to the solvent (Fig. 4b,c). In fact, although all putative oxygen reaction centers previously described for NOXs

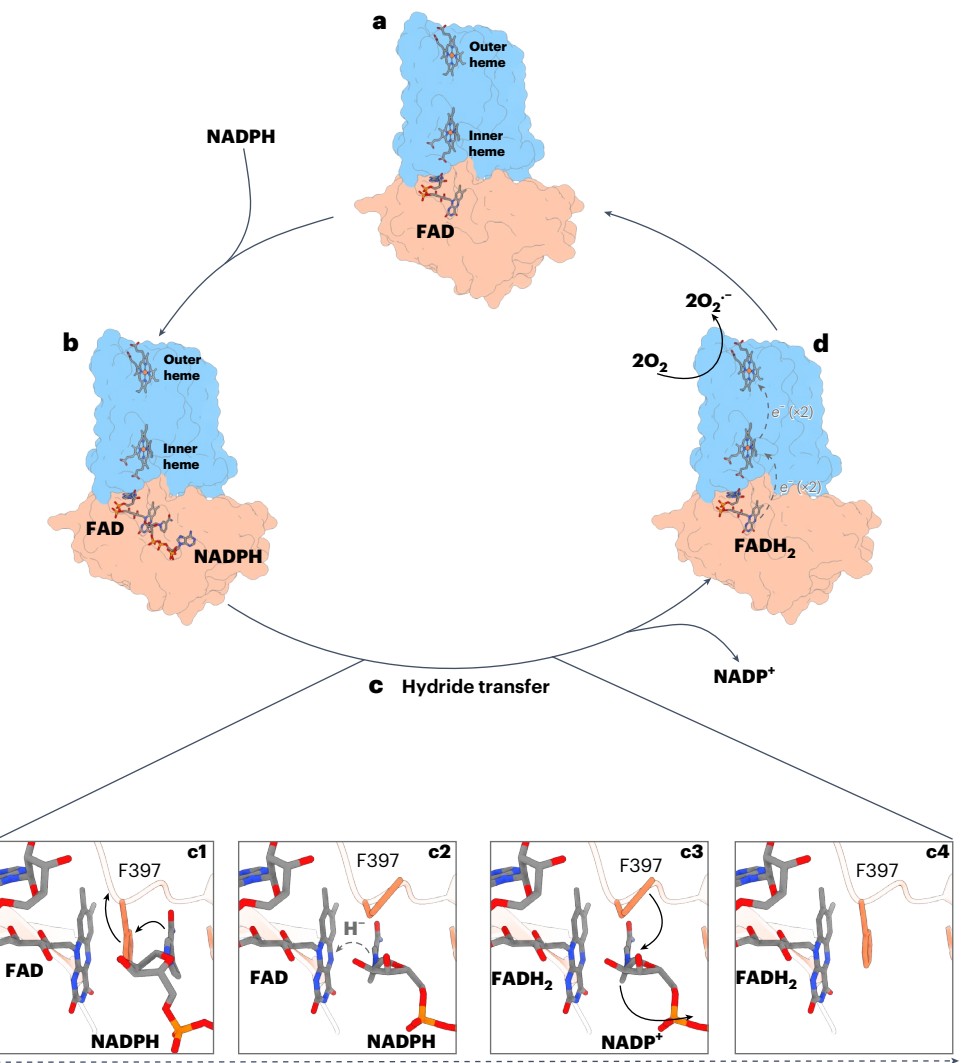

**Fig. 5 | Schematic view of the proposed catalytic cycle of SpNOX and its mechanism of hydride transfer. a–d**, Overall schematic view of SpNOX catalytic cycle. FAD-bound SpNOX is ready for NADPH-binding (**a**). Once the nicotinamide ring interacts with F397 (**b**), the enzyme can catalyze hydride transfer (**c**), which can be divided into several simplified steps: a conformational change involving, at least, the partial displacement of F397 (**c1**); the movement of the nicotinamide ring close enough to the isoalloxazine ring of FAD to perform hydride transfer and generate reduced FAD (FADH₂) (**c2**); in unknown order, the exit of NADP⁺ and the rearrangement of F397 (**c3**) to its original position (**c4**). Then, FADH₂ transfers single electrons through the TMD-bound hemes to an acceptor molecule (in this case, two molecules of O₂) (**d**). Finally, SpNOX goes back to the resting state (**a**). Importantly, although here depicted after **c4** for simplicity, electron transfer (**d**) may happen at any time after **c2**. Dashed gray arrows indicate electron or hydride transfers. Black arrows in **c** indicate amino acid rearrangements.

showed a similar conformation involving two histidine residues, an arginine and a propionate group of the outer heme[8–11], we did not find any similar O₂-binding site in SpNOX. However, a closer examination of the outer heme-binding pocket revealed a small cavity below the C-loop occupied by an ordered water molecule. This water molecule, which forms H-bonds with N84, N101 and Y105, could be occupying an O₂-binding site within efficient electron-transfer distance to the outer heme (Fig. 4d). A potential path for O₂ or O₂•⁻ entry and exit at this site was found using Hollow[27], taking the modeled water molecule as the starting point (Supplementary Fig. 13g). However, N84A and Y105F mutants did not display a notable change in cytochrome c reduction or NADPH oxidation activity compared to WT (Supplementary Fig. 14h–j), suggesting that this site may not be an actual oxygen-reduction center. We also identified another putative O₂-binding site composed of S86 and heme-coordinating H83. Both amino acids are highly exposed to the solvent and coordinate a water molecule that could be occupying the O₂-binding position (Fig. 4e).

## Discussion

Phylogenetic analyses of the FRD-containing superfamily of proteins indicate that the NOX family emerged early in evolution through the fusion of a bacterial FRD-containing protein with an FNR protein[5]. This hypothesis is supported by the recent discovery of NOX-like homologs in bacteria and the biochemical characterization of a member of this new group, SpNOX[4,6]. The cryo-EM structures of this protein presented here provide further relevant details about its catalytic mechanism and add new insights into the electron-transfer pathway of NOXs.

*S. pneumoniae* is a Gram-positive, aerotolerant anaerobic bacterium that is part of the human nasopharyngeal microbiota but can cause serious infections[28]. The mechanisms by which it deals with oxygen and ROS have been extensively studied[29–32]. Remarkably, *S. pneumoniae* is able to produce large amounts of H₂O₂ that appear to promote infection by damaging epithelial cells and other microorganisms of the respiratory tract[33,34]. Nevertheless, contrary to eukaryotic cells, which tightly regulate ROS production by NOXs, *S. pneumoniae* produces H₂O₂ as a metabolic byproduct of enzymes such as the pyruvate oxidase SpxB[30].

Whether bacteria use SpNOX-like proteins for dedicated ROS production in vivo is unresolved. An SpNOX-knockout strain of *S. pneumoniae* was reported to display no discernible phenotype in pure culture[4].

An important characteristic of SpNOX is its constitutive activity in vitro, a feature only shared by mammalian NOX4 homologs. The exact structural basis for this feature remains unknown but published cryo-EM structures of eukaryotic NOX show a highly flexible DH domain whose conformational state is believed to respond to regulatory signals (for example, calcium in DUOX1 or cytosolic factors in NOX2)[9,11]. The structures we obtained suggest that, in SpNOX, the DH domain forms very stable contacts with the TM domain even in the absence of substrate or native lipids, as deduced by the high resolution achieved for both DH and TM domains in a consensus refinement. Therefore, the stability of the interaction of the DH and the TM domains could be a major mechanism to achieve constitutive activity in NOX proteins. Both our substrate-bound structures reveal an unproductive conformation for hydride transfer, providing important detail of the catalytic mechanism of SpNOX. As for other members of the FNR superfamily, a transient displacement of a C-terminal aromatic residue is necessary to perform hydride transfer[24,25]. This is supported by an increase of the turnover rate in an SpNOX F397A mutant, for which our structure shows a productive conformation of the nicotinamide (Fig. 3d,e and Supplementary Fig. 12). Molecular dynamics simulations similar to those already described for *Anabaena* FNR Y303 (refs. 16,35) would aid in our understanding of F397 displacement in SpNOX[16,35], which is disfavored for F397W mutants, resulting in inactivation[12]. A schematic view of the proposed SpNOX catalytic cycle is depicted in Fig. 5. Interestingly, a similar catalytic role was proposed for a strictly conserved phenylalanine residue at the C terminus of eukaryotic NOXs[21]. However, none of the published structures could show this residue interacting with FAD. In csNOX5, the hyperstabilizing substitution used for crystallization of the DH domain seems to prevent F693 from folding natively[7]. In inactive NOX2 complexes, the C-terminal residues could not be resolved because of flexibility[10,11]. In the high-calcium state structure of human DUOX1, F1551 does not interact with FAD and adopts a different conformation far away from the nicotinamide ring. Nevertheless, the distance between the nicotinamide and the isoalloxazine rings is longer than 10 Å, indicating that DUOX1 activation could involve further conformational changes[9].

Despite the large evolutionary distance between eukaryotic NOXs and SpNOX-like bacterial NOXs, our structural and biochemical analyses of SpNOX reveal a highly similar domain organization and fold. This indicates that preserved structural motifs such as the hydrophobic cleft between the inner and the outer heme could be essential for efficient function of the complex. SpNOX also displays relevant specific features not present in eukaryotic NOXs. For example, it lacks the canonical amino acids that configure the potential oxygen-reduction center in the available structures of eukaryotic and cyanobacterial NOXs. Structural analysis revealed two putative oxygen-reduction centers (Fig. 4d,e) with an unusual configuration when compared to other reduction centers previously described in flavoenzymes with oxidase activity[36]. These amino acids are not strictly conserved in other putative bacterial NOXs (Supplementary Fig. 11) but a similar amino acid configuration to the putative center formed by N84, N101 and Y105 could be found in several SpNOX-like proteins by analysis of their structures predicted by AlphaFold 2 (AF2)[37] (Supplementary Fig. 15). However, mutational analysis of these residues did not show a reduction in the activity (Supplementary Fig. 14h–j), indicating that NOX activity of SpNOX could be independent of this site. Therefore, we cannot exclude that oxygen is not the physiological electron acceptor of bacterial SpNOX homologs. The highly exposed outer heme of SpNOX is a feature shared with the protein-methionine-sulfoxide reductase heme-binding subunit MsrQ[38]. MsrQ is considered an ancestor of the FRD superfamily and is part of the MsrPQ system of Gram-negative bacteria[5], transferring electrons from the quinone pool to the periplasmic subunit MsrP, which finally reduces methionine sulfoxide residues of periplasmic proteins. Although the SpNOX TM domain shares only 22% identity with *E. coli* MsrQ, a comparison between the *E. coli* MsrQ AF2 model and substrate-free SpNOX revealed a similar structure (Supplementary Fig. 16). Remarkably, the extracellular region of MsrQ, which is the interaction site of the electron acceptor MsrP, is solvent exposed at the heme-binding cleft. The fact that SpNOX shows an even more solvent-exposed extracellular region (Fig. 4b,c and Supplementary Fig. 16c–e) could indicate that a potential protein acceptor exists for this protein. This would also be supported by the fact that SpNOX can directly reduce cytochrome c and SOD efficiently, as deduced by our (an)aerobic activity assays (Supplementary Fig. 1e,f). Another relevant characteristic of SpNOX is its lack of specificity toward NADPH or NADH. Further functional assays are needed to identify the physiological role of this protein and other bacterial NOX homologs.

A major achievement of our study is the determination of high-resolution cryo-EM structures of SpNOX despite its relatively small size (46 kDa). The fact that we were able to obtain high-resolution reconstructions of such a small membrane protein without the use of a fiducial (for example, a protein-specific nanobody) illustrates the impact of ongoing developments in microscope, detector and software capabilities that continue to push the boundaries of what is possible by single-particle cryo-EM. Important factors in our result are likely to include the production of a highly stable and homogeneous sample (Supplementary Fig. 1a,b), extensive sample optimization for optimal particle distribution in thin ice (Supplementary Figs. 2–5), the use of a Krios G4 microscope with a cold field-emission gun for data collection, a highly stable and low-aberration energy filter[39] and a Falcon 4 detector with high detective quantum efficiency[40], the use of holey gold grids to reduce beam-induced motion[41] and a partially rigid structure that allows particle alignment.

These high-resolution structures, in substrate-free and stably reduced forms and under turnover conditions, reveal important new detail about the catalytic activity and the electron-transfer pathway of this enzyme, shedding light on bacterial NOX homologs and similarities and differences to eukaryotic NOXs. A complete structural understanding of this emerging model enzyme can inform functional studies and drug-discovery experiments involving NOX proteins.

*Note added in proof:* After submission of the revised version of this article, a structure of the activated-state human NOX2 complex was published in which the C-terminal F570 lies between FAD and NADPH, supporting a conserved role for this residue[42].

## Online content

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

## Methods

### Protein expression and purification

An *E. coli* codon-optimized sequence encoding the *S. pneumoniae* R6 strain NOX (2-400, Q8CZ28_STRR6), with an N-terminal 6xHis-tag followed by a PreScission cleavage site, was purchased from GenScript and subcloned between the NcoI and EcoRI sites of the pET-28a plasmid. For the generation of mutants, primers with overlapping sequences (Supplementary Tables 1 and 2) were designed to generate point mutations with the NEBuilder HiFi DNA Assembly mix (New England Biolabs). WT protein expression was performed in OverExpress C41(DE3) *E. coli* cells. Cultures grown in Terrific broth at 37 °C were induced in the exponential phase (optical density at 600 nm, 1.0) by the addition of 0.2 mM IPTG. The cultures were supplemented with 0.5 mM δ-aminolevulinic acid (TCI) to maximize heme incorporation. After 4 h at 37 °C and 140 r.p.m., the cells were harvested and flash-frozen in liquid $N_2$ for later use. For protein purification, the frozen cell pellets were thawed on ice and resuspended in lysis buffer containing 50 mM Tris-HCl pH 7.0 and 300 mM NaCl supplemented with bovine DNase II, 1 mM PMSF and 5 µg ml$^{-1}$ aprotinin, leupeptin and pepstatin (10 ml per 1-g cell pellet) at room temperature for 30 min. Resuspended cells were lysed on ice using an ultrasonic homogenizer (Bandelin) with three cycles at 40% amplitude with 1 s on and 1 s off for a total of 6 min. All subsequent steps were performed at 4 °C. The cell debris was pelleted by centrifugation at 12,108g for 30 min. The membranes were isolated from the supernatant by ultracentrifugation at 185,511g for 1 h. Then, membrane pellets were resuspended in a Dounce homogenizer in lysis buffer. Once resuspended, the sample was supplemented with 0.4% LMNG and gently stirred overnight for solubilization. Nonsolubilized material was removed the next day by ultracentrifugation at 185,511g for 30 min. The supernatant was mixed with Ni-NTA resin equilibrated in lysis buffer and incubated for 2 h while gently rocking. The incubated resin was then loaded onto a gravity column and washed with ten column volumes of lysis buffer supplemented with 5 mM imidazole and 0.002% LMNG. The protein was eluted with 300 mM imidazole and immediately diluted 1:1 with imidazole-free lysis buffer supplemented with 0.002% LMNG to avoid aggregation. PD-10 columns (GE Healthcare) were used for buffer exchange to 50 mM Tris-HCl pH 7.5, 150 mM NaCl and 0.002% LMNG. His-PreScission protease, expressed and purified in house[43], was added to a final concentration of 0.05 mg ml$^{-1}$ and incubated overnight. The sample was centrifuged at 12,108g for 30 min to pellet aggregates, supplemented with 20 mM imidazole and loaded onto a Ni-NTA gravity column equilibrated in 50 mM Tris-HCl pH 7.5, 150 mM NaCl, 20 mM imidazole and 0.002% LMNG. The flow-through volume containing the untagged protein was concentrated to 10 mg ml$^{-1}$ using Vivacon500 filters (50 kDa, Sartorius) for further purification using a Superdex 200 Increase 5/150 GL column equilibrated in 50 mM Tris-HCl pH 7.0, 250 mM NaCl and 0.002% LMNG. Fractions containing the monodisperse protein were pooled, concentrated for cryo-EM sample preparation or flash-frozen for future use.

### Cryo-EM sample preparation and data collection

All cryo-EM grids were prepared from freshly purified SpNOX concentrated to 6 mg ml$^{-1}$ for substrate-free samples and 7.5 mg ml$^{-1}$ for substrate-present samples and supplemented with 1 mM FAD at least 30 min before grid freezing.

For the substrate-free structure, 3 µl of SpNOX was directly applied to freshly glow-discharged R 1.2/1.3 400-mesh UltrAuFoil grids. The grids were blotted using Whatman 595 blotting paper (Sigma-Aldrich) for 4 to 6 s at 4 °C and 100% humidity and then plunge-frozen into liquid ethane using a Vitrobot Mark IV (Thermo Fisher Scientific). A dataset was collected using a Titan Krios G4 equipped with a cold field-emission gun, Selectris X filter and Falcon 4 direct electron detector. EPU (version 3.2) was used for automated data acquisition of 18,433 movies in electron event representation (EER) format[44]. Movies were collected with a pixel size of 0.573 Å and a total dose of 70 e$^-$ per Å$^2$ contained in 875 EER frames. The defocus ranged from −0.7 to −2.1 µm.

For the NADPH-bound structure, grids were prepared as for the substrate-free SpNOX except that 2.7 µl of SpNOX was mixed with 0.3 µl of 10 mM NADPH and immediately applied to glow-discharged R 1.2/1.3 400-mesh UltrAuFoil grids inside the Vitrobot chamber. The amount of time from mixing to plunge-freezing was not longer than 15 s. A dataset was collected using the same microscope and imaging settings as for the substrate-free sample (EPU version 3.3). The dataset contained 22,259 movies with 1,036 EER frames.

For the NADH-bound structure, an aliquot of aerobically purified SpNOX was put inside an anaerobic chamber (<1 ppm $O_2$; Coy Laboratory Products) and mixed with 5 mM NADH from a 50 mM NADH stock. After 40 min, 3 µl of the sample was applied to glow-discharged R 1.2/1.3 400-mesh UltrAuFoil grids, blotted for 4–5 s at 4 °C under 100% humidity and plunge-frozen in liquid ethane using a Vitrobot Mark IV (Thermo Fisher Scientific). A dataset was collected using the same microscope and imaging settings as for the aerobically frozen samples (EPU version 3.3). The dataset contained 28,143 movies with 938 EER frames.

For the F397A mutant structure, grids were prepared as for the NADPH-bound structure. First, 2.7 µl of SpNOX was mixed with 0.3 µl of 50 mM NADPH and immediately applied to glow-discharged R 1.2/1.3 400-mesh UltrAuFoil grids inside the Vitrobot chamber. The grids were blotted for 5–6 s and plunge-frozen in liquid ethane. Four datasets were collected on two grids using the same microscope and imaging settings as for the other datasets (EPU version 3.4). Plasmon imaging was used to efficiently select holes with the optimal ice thickness[45]. The datasets contained 21,851, 9,538, 6,332 and 7,415 movies with 924, 899, 1,008 and 994 EER frames, respectively.

### Image processing

Detailed workflows for each dataset are shown in Supplementary Figs. 2–5. Processing of the substrate-free dataset started by importing all movies into RELION-4.0 (ref. 46) and the raw movies were fractioned to 1.04 e$^-$ per Å$^2$ per frame, motion-corrected and dose-weighted using RELION's implementation of MotionCor2 with 5 × 5 patches[47]. CTF parameters were estimated with CTFFIND4.1 (ref. 48). Micrographs were then selected on the basis of a maximum 4.0-Å contrast transfer function (CTF) resolution estimate and a defocus value between −0.5 and −2.5 µm. For particle picking, different programs and models were used in parallel. After classification and removal of 'junk' particles, the particle sets from the different strategies were merged and duplicates were removed. Particles were picked with two trained Topaz models[49], as well as a refined standard crYOLO[50] model, from all motion-corrected micrographs. The Topaz models were trained on a previous dataset that did not yield a high-resolution reconstruction. The crYOLO model was refined with ~800 manually picked particles from selected micrographs of the substrate-free dataset. A picking threshold of 0.1 was used for crYOLO picking. Picked particles were extracted with a 256-pixel box size and downsampled to 56 pixels. Topaz particles were extracted with a figure of merit (FOM) threshold of −1.5. The particles were imported into cryoSPARC (version 3.2)[51] and subjected separately to several rounds of two-dimensional (2D) classification. Initial models were generated from 1,000,000 particles of the crYOLO particle set. The initial models were used as references for a heterogenous refinement for each particle set with three references. The classes representing the protein were transferred to RELION format before being merged to remove duplicate particles, resulting in a 1,160,039-particle dataset that was re-extracted in a box of 320 pixels downsampled to 72 pixels. Particle transfer between RELION and cryoSPARC was performed by the UCSF pyem program package[52]. Several rounds of nonuniform (NU) refinement and re-extraction in a larger particle box with less downsampling were performed followed by local and global CTF refinement that resulted in a 2.65-Å reconstruction using a 448-pixel box downsampled to 256 pixels. A large extraction box was necessary to capture all the delocalized signal of the protein particle. Several rounds of heterogenous refinement, CTF refinement and RELION-assisted particle

polishing[53] resulted in an NU refinement resolved to 2.40 Å. A final local refinement with a tighter mask resulted in a 2.29-Å final reconstruction from 379,972 particles sharpened with a B factor of −80.1 Å$^2$.

Processing of the turnover conditions with NADPH present started by importing all movies into RELION-4.0 and the movies were fractioned to 1.04 e$^-$ per Å$^2$ per frame, motion-corrected and dose-weighted using RELION's implementation of MotionCor2 with 5 × 5 patches. Subsequently, CTF parameters were estimated using CTFFIND4.1 and micrographs were selected on the basis of a maximum of 3.5-Å CTF resolution estimate and a defocus value between −0.5 and −2.5 µm. The same three particle picking models and parameters were used as for the substrate-free dataset. The particles were extracted in a 256-pixel box and downsampled to 56 pixels. For the Topaz picked particles, an FOM threshold of −1.5 was used. The particles were imported into cryoSPARC (version 4.0) and subjected to two rounds of 2D classification. One of the three particle sets was used to generate four initial models, of which three were used for subsequent heterogenous refinement for each particle set after 2D classification. The SpNOX-representing classes were transferred to RELION using the UCSF pyem program package. The particles were merged and duplicates removed before re-extraction in a larger box with smaller downsampling. The particles were then imported into cryoSPARC and another ab initio step was performed to classify the particles into two classes before several rounds of NU and CTF refinements with increasingly larger extraction box size and smaller downsampling. RELION-assisted particle polishing and several iterative rounds of heterogenous refinement resulted in an NU refinement reconstruction reaching 2.25 Å using 591,137 particles in a 448-pixel box downsampled to 280 pixels and a loose mask including the detergent micelle. A subsequent local refinement with a tight mask excluding the detergent micelle resulted in the final reconstruction reaching 2.20-Å nominal resolution sharpened with a B factor of −74.8 Å$^2$.

Processing of the dataset under stably reducing conditions with NADH started by importing the raw movies into RELION-4.0. The movies were fractioned to 1.04 e$^-$ per Å$^2$ per frame, motion-corrected using 5 × 5 patches and dose-weighted using RELION's implementation of MotionCor2. The CTF parameters were estimated using CTFFIND4.1. Micrographs with a maximum of 3.5-Å CTF resolution estimate and a defocus value between −0.5 and −2.5 µm were selected. The same particle picking models as for the other datasets were used. Particles were extracted in a 256-pixel box and downsampled to 56 pixels. Topaz particles were extracted with an FOM threshold of −1.5. The particles were transferred to cryoSPARC (version 4.0) for 2–3 rounds of 2D classification. Using one particle set, three initial models were generated and these were used for a heterogenous refinement of all three particles sets independently. The SpNOX representing classes were transferred to RELION using the UCSF pyem program package. The particle sets were merged and duplicate particles were removed. The resulting particle set was re-extracted with a larger box size and less downsampling and then transferred to cryoSPARC. Several rounds of heterogenous, NU and CTF refinements, multiple re-extractions in a larger box with progressively less downsampling and RELION-assisted particle polishing resulted in a final particle set of 697,514 particles. The final NU refinement with a loose mask resulted in a 2.42-Å reconstruction and the subsequent local refinement resulted in a 2.36-Å nominal resolution reconstruction sharpened with a B factor of −79.9 Å$^2$.

Processing of the four F397A mutant datasets started with the import of the raw movies into RELION-4.0 using separate optics groups. The raw movies were motion-corrected and fractioned to 0.98, 1.03, 0.972 and 0.98 e$^-$ per Å$^2$ per frame, respectively, and dose-weighted using RELION's implementation of MotionCor2. The CTF parameters of the micrographs were estimated using CTFFIND4.1. A selection of micrographs was made on the basis of 4-Å maximum CTF resolution and defocus values between −0.5 and −2.5 µm. The same particle picking models were used as for the other datasets. Particles were extracted using a 256-pixel box downsampled to 56 pixels. The particles picked using Topaz were extracted with an FOM of −1.5. All particles from different datasets picked using the same model were merged. The particle sets were imported into cryoSPARC (version 4.0) for two rounds of 2D classification. Three initial models were generated and used as references for a round of heterogeneous refinement. The SpNOX-representing classes were reimported into RELION and merged before duplicate particles were removed. The resulting particle set was subjected to iterative rounds of NU local refinement, CTF refinement, heterogeneous refinement, re-extraction into larger boxes with less downsampling and RELION-assisted particle polishing. The final particle set contained 546,234 particles. The final NU refinement with a loose mask including the detergent micelle reached 2.73 Å and a subsequent local refinement using a mask excluding the detergent micelle resulted in a 2.64-Å nominal resolution reconstruction sharpened with a B factor of −105.4 Å$^2$.

## Model building

For the SpNOX apo model, the structure predicted by AF2 (ref. 37) was used as an initial model. UCSF ChimeraX (version 1.7)[54] was used to rigidly fit the model in the density map. The program Coot[55] was used to place the cofactors and to inspect and adjust all coordinates manually. Several iterative rounds of Phenix (version 1.20) real-space refinement[56] and manual adjustment in Coot (version 0.9) were performed until the stereochemistry was good and fit the cryo-EM density map as assessed with Phenix, MolProbity (version 4.5)[57] and Q-score[58].

The substrate-bound models were built using the substrate-free model as an initial model except for the F397A model, where the NADPH model was used as an initial model. The respective substrates were fit into the density using Coot. Several iterations of Phenix real-space refinement and manual adjustment were performed until the stereochemistry and fit in the density map were satisfactory as assessed by Phenix, MolProbity and Q-score.

For structural comparison, models were aligned using the Matchmaker tool in UCSF ChimeraX. Cavities were calculated and drawn using Hollow (version 1.3)[27] with a 1.1-Å probe radius. Finalized models were visualized using UCSF ChimeraX. Electrostatic potential was calculated using the adaptive Poisson−Boltzmann solver tool (version 3.4.1)[59] and visualized in UCSF ChimeraX. Search of bacterial NOX-like proteins was performed using the basic local alignment similarity search tool in UniProt[60] against the UniProtKB and Swiss-Prot databases with default settings and bacteria (eubacteria) as the taxonomy restrictor. AF2 models were downloaded from the AF Protein Structure Database (AF DB)[37,61]. Sequence alignments were performed with PROMALS3D (ref. 62) and visualized using the ENDscript server[63]. Sequence identity and similarity (using GAVLI, FYW, CM, ST, KRH, DENQ and P as default similarity groups) were calculated using the Sequence Manipulation Suite (version 2)[64].

## Activity assays

The NOX activity of SpNOX was measured using the cytochrome c reduction assay as previously described[65], with some modifications. The reaction was performed at room temperature in a final volume of 0.5 ml in 50 mM Tris pH 7.0 and 250 mM NaCl with 100 µM bovine heart cytochrome c (Sigma). Before the assay, the solution was supplemented with 0.1 µM SpNOX and 1 µM FAD from 50 µM and 10 mM stocks, respectively. The reaction was triggered by the addition of NADPH or NADH from 25 mM stocks and the reduction was followed by measuring the absorbance at 550 nm using a spectrophotometer (Varian Cary 50). When indicated, bovine SOD (recombinant; Sigma) was added from a ≥50,000 U per ml stock prepared in assay buffer. Anaerobic cytochrome c and SOD reduction was measured with 0.05 µM SpNOX using a Nanodrop One with cuvette holder (Thermo Fisher Scientific) inside an anaerobic chamber (<1 ppm O$_2$; Coy Laboratory Products). Cytochrome c was added to a final concentration of 100 µM and SOD

was added to a final concentration of 1,000 U per ml. The oxidation of NADPH and reduction of cytochrome c were followed by measuring the absorbance at 340 and 550 nm, respectively. Aerobic and anaerobic NADPH oxidation was measured using 0.5 μM SpNOX and the reaction was initiated by the addition of 100 μM NADPH. The comparison of aerobic and anaerobic cytochrome c reduction was measured using 0.5 μM SpNOX, 100 μM NADPH and 100 μM cytochrome c. Iron reductase assays were performed as previously described[66], with some modifications. A standard curve for the ferrozine absorbance coefficient was generated using $Fe^{II}Cl_2$ (Sigma) for each assay. The anaerobic assay was performed at room temperature in a final volume of 1 ml in 50 mM Tris pH 7.0, 250 mM NaCl. The sample was supplemented with 0.5 μM SpNOX, 800 μM ferrozine (Sigma), 100 μM $Fe^{III}$-EDTA (Sigma) and 5 μM FAD. The reaction was initiated by the addition of 100 μM NADPH. The reduction of $Fe^{III}$ to $Fe^{II}$ was followed at 562 nm using a Nanodrop One with cuvette holder. Michaelis–Menten kinetics for $Fe^{III}$ reduction were measured at room temperature in a final volume of 0.5 ml with 50 mM Tris pH 7.0 and 250 mM NaCl with 800 μM ferrozine, supplemented with 0.5 μM SpNOX, 1 μM FAD and varying $Fe^{III}$-EDTA concentrations. The reaction was initiated by the addition of 200 μM NADPH and the absorbance at 562 nm was followed using a spectrophotometer. Data were analyzed and fit using OriginLab.

## Reporting summary

Further information on research design is available in the Nature Portfolio Reporting Summary linked to this article.

## Data availability

Atomic models and maps for substrate-free SpNOX were deposited to the Protein Data Bank (PDB) with accession code 8QT6 and EM Data Bank with accession code EMD-18644; for anaerobically frozen NADH-bound SpNOX, the model and map are available as 8QT9 and EMD-18646; for NADPH-bound SpNOX under turnover conditions, the model and map are available as 8QT7 and EMD-18645; for NADPH-bound F397A SpNOX under turnover conditions, the model and map are available as 8QTA and EMD-18647. Other data and materials are available from the corresponding authors upon request. Raw data were deposited to EMPIAR under the following accession codes: substrate-free SpNOX, 11741; NADH-bound SpNOX, 11743; NADPH-bound SpNOX, 11742; NADPH-bound F397A SpNOX, 11744. Sequence alignments were performed against the UniProtKB and Swiss-Prot databases, which are publicly available. The entries 5OOT, 5OOX, 8GZ3, 7D3F and 1QUF used in this study were downloaded from the PDB. Source data are provided with this paper.

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

## Acknowledgements

We thank the Central EM Facility of the Max Planck Institute of Biophysics for providing cryo-EM infrastructure and technical support and we thank W. Kühlbrandt and the Department of Structural Biology for support and helpful discussions. We thank R. Zimmermann for sharing her valuable expertise in anaerobic work. This work was funded by the Max Planck Society (to B.J.M.) and an Alfonso Martín Escudero postdoctoral fellowship (to P.S.S-A.).

## Author contributions

P.S.S.-A. and B.J.M. conceptualized the project. P.S.S.-A. and V.R.A.D. performed the biochemical and structural experiments and analyzed the data. P.S.S.-A., V.R.A.D and B.J.M. wrote the paper. B.J.M. supervised the project.

## Funding

## Competing interests

The authors declare no competing interests.

## Additional information

**Correspondence and requests for materials** should be addressed to Pablo San Segundo-Acosta or Bonnie J. Murphy.

# Reporting Summary

## Statistics

For all statistical analyses, confirm that the following items are present in the figure legend, table legend, main text, or Methods section.

| n/a | Confirmed | |
|---|---|---|
| ☐ | ☒ | The exact sample size (*n*) for each experimental group/condition, given as a discrete number and unit of measurement |
| ☐ | ☒ | A statement on whether measurements were taken from distinct samples or whether the same sample was measured repeatedly |
| ☒ | ☐ | The statistical test(s) used AND whether they are one- or two-sided<br>*Only common tests should be described solely by name; describe more complex techniques in the Methods section.* |
| ☒ | ☐ | A description of all covariates tested |
| ☒ | ☐ | A description of any assumptions or corrections, such as tests of normality and adjustment for multiple comparisons |
| ☐ | ☒ | A full description of the statistical parameters including central tendency (e.g. means) or other basic estimates (e.g. regression coefficient) AND variation (e.g. standard deviation) or associated estimates of uncertainty (e.g. confidence intervals) |
| ☒ | ☐ | For null hypothesis testing, the test statistic (e.g. $F$, $t$, $r$) with confidence intervals, effect sizes, degrees of freedom and $P$ value noted<br>*Give P values as exact values whenever suitable.* |
| ☒ | ☐ | For Bayesian analysis, information on the choice of priors and Markov chain Monte Carlo settings |
| ☒ | ☐ | For hierarchical and complex designs, identification of the appropriate level for tests and full reporting of outcomes |
| ☒ | ☐ | Estimates of effect sizes (e.g. Cohen's *d*, Pearson's *r*), indicating how they were calculated |

*Our web collection on statistics for biologists contains articles on many of the points above.*

## Software and code

Policy information about availability of computer code

| Data collection | EPU v3.2, v3.3, v3.4 (Thermo Fisher Scientific) |
|---|---|
| Data analysis | cryoSPARC v3.2 and 4.1; Relion 4.0; CTFFind4.1; crYOLO v1.8.4; pyem v0.5; Phenix v1.20; Coot 0.9; MolProbity v4.5; ChimeraX v1.7; Hollow v1.3; OriginLab v.2022; Adaptive Poisson-Boltzmann Solver (APBS) v3.4.1; UniProt BLAST; PROMALS3D; Sequence_Manipulation_Suite v2. |

For manuscripts utilizing custom algorithms or software that are central to the research but not yet described in published literature, software must be made available to editors and reviewers. We strongly encourage code deposition in a community repository (e.g. GitHub). See the Nature Portfolio guidelines for submitting code & software for further information.

## Data

Policy information about availability of data

All manuscripts must include a data availability statement. This statement should provide the following information, where applicable:
- Accession codes, unique identifiers, or web links for publicly available datasets
- A description of any restrictions on data availability
- For clinical datasets or third party data, please ensure that the statement adheres to our policy

Atomic models and maps for substrate-free SpNOX were deposited in the Protein Data Bank with accession code 8QT6 and Electron Microscopy Data Bank with accession code EMD-18644; for anaerobically frozen NADH-bound SpNOX the model and map are available as 8QT9 and EMD-18646; for NADPH-bound SpNOX under turnover conditions the model and map are available as 8QT7 and EMD-18645; and for NADPH-bound F397A SpNOX under turnover conditions the model

and map are available as 8QTA and EMD-18647. Other data and materials are available from the corresponding authors upon request. Raw data have been deposited in EMPIAR under the following accession codes: substrate-free SpNOX (11741), NADH-bound (11743), NADPH-bound SpNOX (11742) and NADPH-bound F397A SpNOX (11744). Sequence alignments were performed against the UniProtKB and Swiss-Prot databases which are publicly available. Source data for raw images of cropped gels, as well as individual replicates of all assay results, are provided with this paper. The entries 5O0T, 5O0X, 8GZ3, 7D3F and 1QUF used in this study were downloaded from the PDB.

# Research involving human participants, their data, or biological material

Policy information about studies with human participants or human data. See also policy information about sex, gender (identity/presentation), and sexual orientation and race, ethnicity and racism.

| | |
|---|---|
| Reporting on sex and gender | N/A |
| Reporting on race, ethnicity, or other socially relevant groupings | N/A |
| Population characteristics | N/A |
| Recruitment | N/A |
| Ethics oversight | N/A |

Note that full information on the approval of the study protocol must also be provided in the manuscript.

# Field-specific reporting

Please select the one below that is the best fit for your research. If you are not sure, read the appropriate sections before making your selection.

☒ Life sciences   ☐ Behavioural & social sciences   ☐ Ecological, evolutionary & environmental sciences

For a reference copy of the document with all sections, see nature.com/documents/nr-reporting-summary-flat.pdf

# Life sciences study design

All studies must disclose on these points even when the disclosure is negative.

| | |
|---|---|
| Sample size | For structural data, the number of micrographs collected was determined by available microscope time. For enzyme assays, a standard of 3 replicates were performed, which gave reproducible results. |
| Data exclusions | In the case of the cytochrome c reduction assay of wild-type spNOX at 20μM NADH, a single replicate gave strongly divergent results indicating likely experimental error; the result of this replicate was excluded. In the revised version of the manuscript, values indicating sequence identity and similarity were re-calculated from the alignments obtained with the advanced alignment tool PROMALS3D, which were more accurate as judged by the correspondence to structural alignments. Therefore, previous values calculated using ClustalOmega alignments were excluded from the manuscript. |
| Replication | Enzyme assays were conducted in triplicate (technical replicates); in the case of the cytochrome c reduction assay of wild-type spNOX at 20μM NADH, a single replicate gave strongly divergent results; for this condition, an additional replicate was therefore performed. In the revised version of the manuscript, for the Y353R SpNOX mutant, we noticed that a single replicate gave strongly divergent results; three more technical replicates were performed for each data point to get a more accurate kinetic analysis for this mutant, using the same batch of recombinant enzyme. Apart from the above-mentioned, all enzymatic assays replicates were successful. Protein purification for WT SpNOX was performed more than 3 times which were all successful. Protein purification for the mutants was done up to 2 times which yielded enough protein to perform the enzymatic assays and determine the structure of the F397A mutant. Cryo-EM datasets were typically collected on a single grid, except for the F397A structure, which was determined using images from 2 grids. |
| Randomization | For cryo-EM, particles were automatically randomized between data half-sets by the refinement software. No group allocation was needed for functional experiments in this study. |
| Blinding | Blinding was not carried out for this study. Experimenter intervention is required for successful single-particle analysis at the class selection stage, which is an established practice in the field. For biochemical assays, no blinding is carried out as results are collected without subjective experimenter intervention. |

# Behavioural & social sciences study design

All studies must disclose on these points even when the disclosure is negative.

| | |
|---|---|
| Study description | *Briefly describe the study type including whether data are quantitative, qualitative, or mixed-methods (e.g. qualitative cross-sectional, quantitative experimental, mixed-methods case study).* |

| Research sample | *State the research sample (e.g. Harvard university undergraduates, villagers in rural India) and provide relevant demographic information (e.g. age, sex) and indicate whether the sample is representative. Provide a rationale for the study sample chosen. For studies involving existing datasets, please describe the dataset and source.* |
|---|---|
| Sampling strategy | *Describe the sampling procedure (e.g. random, snowball, stratified, convenience). Describe the statistical methods that were used to predetermine sample size OR if no sample-size calculation was performed, describe how sample sizes were chosen and provide a rationale for why these sample sizes are sufficient. For qualitative data, please indicate whether data saturation was considered, and what criteria were used to decide that no further sampling was needed.* |
| Data collection | *Provide details about the data collection procedure, including the instruments or devices used to record the data (e.g. pen and paper, computer, eye tracker, video or audio equipment) whether anyone was present besides the participant(s) and the researcher, and whether the researcher was blind to experimental condition and/or the study hypothesis during data collection.* |
| Timing | *Indicate the start and stop dates of data collection. If there is a gap between collection periods, state the dates for each sample cohort.* |
| Data exclusions | *If no data were excluded from the analyses, state so OR if data were excluded, provide the exact number of exclusions and the rationale behind them, indicating whether exclusion criteria were pre-established.* |
| Non-participation | *State how many participants dropped out/declined participation and the reason(s) given OR provide response rate OR state that no participants dropped out/declined participation.* |
| Randomization | *If participants were not allocated into experimental groups, state so OR describe how participants were allocated to groups, and if allocation was not random, describe how covariates were controlled.* |

# Ecological, evolutionary & environmental sciences study design

All studies must disclose on these points even when the disclosure is negative.

| Study description | *Briefly describe the study. For quantitative data include treatment factors and interactions, design structure (e.g. factorial, nested, hierarchical), nature and number of experimental units and replicates.* |
|---|---|
| Research sample | *Describe the research sample (e.g. a group of tagged Passer domesticus, all Stenocereus thurberi within Organ Pipe Cactus National Monument), and provide a rationale for the sample choice. When relevant, describe the organism taxa, source, sex, age range and any manipulations. State what population the sample is meant to represent when applicable. For studies involving existing datasets, describe the data and its source.* |
| Sampling strategy | *Note the sampling procedure. Describe the statistical methods that were used to predetermine sample size OR if no sample-size calculation was performed, describe how sample sizes were chosen and provide a rationale for why these sample sizes are sufficient.* |
| Data collection | *Describe the data collection procedure, including who recorded the data and how.* |
| Timing and spatial scale | *Indicate the start and stop dates of data collection, noting the frequency and periodicity of sampling and providing a rationale for these choices. If there is a gap between collection periods, state the dates for each sample cohort. Specify the spatial scale from which the data are taken* |
| Data exclusions | *If no data were excluded from the analyses, state so OR if data were excluded, describe the exclusions and the rationale behind them, indicating whether exclusion criteria were pre-established.* |
| Reproducibility | *Describe the measures taken to verify the reproducibility of experimental findings. For each experiment, note whether any attempts to repeat the experiment failed OR state that all attempts to repeat the experiment were successful.* |
| Randomization | *Describe how samples/organisms/participants were allocated into groups. If allocation was not random, describe how covariates were controlled. If this is not relevant to your study, explain why.* |
| Blinding | *Describe the extent of blinding used during data acquisition and analysis. If blinding was not possible, describe why OR explain why blinding was not relevant to your study.* |

Did the study involve field work? ☐ Yes ☐ No

# Field work, collection and transport

| Field conditions | *Describe the study conditions for field work, providing relevant parameters (e.g. temperature, rainfall).* |
|---|---|
| Location | *State the location of the sampling or experiment, providing relevant parameters (e.g. latitude and longitude, elevation, water depth).* |
| Access & import/export | *Describe the efforts you have made to access habitats and to collect and import/export your samples in a responsible manner and in* |

| | |
|---|---|
| Access & import/export | *compliance with local, national and international laws, noting any permits that were obtained (give the name of the issuing authority, the date of issue, and any identifying information).* |
| Disturbance | *Describe any disturbance caused by the study and how it was minimized.* |

# Reporting for specific materials, systems and methods

We require information from authors about some types of materials, experimental systems and methods used in many studies. Here, indicate whether each material, system or method listed is relevant to your study. If you are not sure if a list item applies to your research, read the appropriate section before selecting a response.

## Materials & experimental systems

| n/a | Involved in the study |
|---|---|
| ☒ ☐ | Antibodies |
| ☒ ☐ | Eukaryotic cell lines |
| ☒ ☐ | Palaeontology and archaeology |
| ☒ ☐ | Animals and other organisms |
| ☒ ☐ | Clinical data |
| ☒ ☐ | Dual use research of concern |
| ☒ ☐ | Plants |

## Methods

| n/a | Involved in the study |
|---|---|
| ☒ ☐ | ChIP-seq |
| ☒ ☐ | Flow cytometry |
| ☒ ☐ | MRI-based neuroimaging |

## Antibodies

| | |
|---|---|
| Antibodies used | *Describe all antibodies used in the study; as applicable, provide supplier name, catalog number, clone name, and lot number.* |
| Validation | *Describe the validation of each primary antibody for the species and application, noting any validation statements on the manufacturer's website, relevant citations, antibody profiles in online databases, or data provided in the manuscript.* |

## Eukaryotic cell lines

Policy information about cell lines and Sex and Gender in Research

| | |
|---|---|
| Cell line source(s) | *State the source of each cell line used and the sex of all primary cell lines and cells derived from human participants or vertebrate models.* |
| Authentication | *Describe the authentication procedures for each cell line used OR declare that none of the cell lines used were authenticated.* |
| Mycoplasma contamination | *Confirm that all cell lines tested negative for mycoplasma contamination OR describe the results of the testing for mycoplasma contamination OR declare that the cell lines were not tested for mycoplasma contamination.* |
| Commonly misidentified lines (See ICLAC register) | *Name any commonly misidentified cell lines used in the study and provide a rationale for their use.* |

## Palaeontology and Archaeology

| | |
|---|---|
| Specimen provenance | *Provide provenance information for specimens and describe permits that were obtained for the work (including the name of the issuing authority, the date of issue, and any identifying information). Permits should encompass collection and, where applicable, export.* |
| Specimen deposition | *Indicate where the specimens have been deposited to permit free access by other researchers.* |
| Dating methods | *If new dates are provided, describe how they were obtained (e.g. collection, storage, sample pretreatment and measurement), where they were obtained (i.e. lab name), the calibration program and the protocol for quality assurance OR state that no new dates are provided.* |

☐ Tick this box to confirm that the raw and calibrated dates are available in the paper or in Supplementary Information.

| | |
|---|---|
| Ethics oversight | *Identify the organization(s) that approved or provided guidance on the study protocol, OR state that no ethical approval or guidance was required and explain why not.* |

Note that full information on the approval of the study protocol must also be provided in the manuscript.

# Animals and other research organisms

Policy information about studies involving animals; ARRIVE guidelines recommended for reporting animal research, and Sex and Gender in Research

| | |
|---|---|
| Laboratory animals | *For laboratory animals, report species, strain and age OR state that the study did not involve laboratory animals.* |
| Wild animals | *Provide details on animals observed in or captured in the field; report species and age where possible. Describe how animals were caught and transported and what happened to captive animals after the study (if killed, explain why and describe method; if released, say where and when) OR state that the study did not involve wild animals.* |
| Reporting on sex | *Indicate if findings apply to only one sex; describe whether sex was considered in study design, methods used for assigning sex. Provide data disaggregated for sex where this information has been collected in the source data as appropriate; provide overall numbers in this Reporting Summary. Please state if this information has not been collected.  Report sex-based analyses where performed, justify reasons for lack of sex-based analysis.* |
| Field-collected samples | *For laboratory work with field-collected samples, describe all relevant parameters such as housing, maintenance, temperature, photoperiod and end-of-experiment protocol OR state that the study did not involve samples collected from the field.* |
| Ethics oversight | *Identify the organization(s) that approved or provided guidance on the study protocol, OR state that no ethical approval or guidance was required and explain why not.* |

Note that full information on the approval of the study protocol must also be provided in the manuscript.

# Clinical data

Policy information about clinical studies
All manuscripts should comply with the ICMJE guidelines for publication of clinical research and a completed CONSORT checklist must be included with all submissions.

| | |
|---|---|
| Clinical trial registration | *Provide the trial registration number from ClinicalTrials.gov or an equivalent agency.* |
| Study protocol | *Note where the full trial protocol can be accessed OR if not available, explain why.* |
| Data collection | *Describe the settings and locales of data collection, noting the time periods of recruitment and data collection.* |
| Outcomes | *Describe how you pre-defined primary and secondary outcome measures and how you assessed these measures.* |

# Dual use research of concern

Policy information about dual use research of concern

## Hazards

Could the accidental, deliberate or reckless misuse of agents or technologies generated in the work, or the application of information presented in the manuscript, pose a threat to:

| No | Yes | |
|----|-----|---|
| ☐ | ☐ | Public health |
| ☐ | ☐ | National security |
| ☐ | ☐ | Crops and/or livestock |
| ☐ | ☐ | Ecosystems |
| ☐ | ☐ | Any other significant area |

## Experiments of concern

Does the work involve any of these experiments of concern:

No  Yes

☐ | ☐ Demonstrate how to render a vaccine ineffective

☐ | ☐ Confer resistance to therapeutically useful antibiotics or antiviral agents

☐ | ☐ Enhance the virulence of a pathogen or render a nonpathogen virulent

☐ | ☐ Increase transmissibility of a pathogen

☐ | ☐ Alter the host range of a pathogen

☐ | ☐ Enable evasion of diagnostic/detection modalities

☐ | ☐ Enable the weaponization of a biological agent or toxin

☐ | ☐ Any other potentially harmful combination of experiments and agents

## Plants

| | |
|---|---|
| Seed stocks | *Report on the source of all seed stocks or other plant material used. If applicable, state the seed stock centre and catalogue number. If plant specimens were collected from the field, describe the collection location, date and sampling procedures.* |
| Novel plant genotypes | *Describe the methods by which all novel plant genotypes were produced. This includes those generated by transgenic approaches, gene editing, chemical/radiation-based mutagenesis and hybridization. For transgenic lines, describe the transformation method, the number of independent lines analyzed and the generation upon which experiments were performed. For gene-edited lines, describe the editor used, the endogenous sequence targeted for editing, the targeting guide RNA sequence (if applicable) and how the editor was applied.* |
| Authentication | *Describe any authentication procedures for each seed stock used or novel genotype generated. Describe any experiments used to assess the effect of a mutation and, where applicable, how potential secondary effects (e.g. second site T-DNA insertions, mosiacism, off-target gene editing) were examined.* |

## ChIP-seq

### Data deposition

☐ Confirm that both raw and final processed data have been deposited in a public database such as GEO.

☐ Confirm that you have deposited or provided access to graph files (e.g. BED files) for the called peaks.

| | |
|---|---|
| Data access links<br>*May remain private before publication.* | *For "Initial submission" or "Revised version" documents, provide reviewer access links. For your "Final submission" document, provide a link to the deposited data.* |
| Files in database submission | *Provide a list of all files available in the database submission.* |
| Genome browser session<br>(e.g. UCSC) | *Provide a link to an anonymized genome browser session for "Initial submission" and "Revised version" documents only, to enable peer review. Write "no longer applicable" for "Final submission" documents.* |

### Methodology

| | |
|---|---|
| Replicates | *Describe the experimental replicates, specifying number, type and replicate agreement.* |
| Sequencing depth | *Describe the sequencing depth for each experiment, providing the total number of reads, uniquely mapped reads, length of reads and whether they were paired- or single-end.* |
| Antibodies | *Describe the antibodies used for the ChIP-seq experiments; as applicable, provide supplier name, catalog number, clone name, and lot number.* |
| Peak calling parameters | *Specify the command line program and parameters used for read mapping and peak calling, including the ChIP, control and index files used.* |
| Data quality | *Describe the methods used to ensure data quality in full detail, including how many peaks are at FDR 5% and above 5-fold enrichment.* |
| Software | *Describe the software used to collect and analyze the ChIP-seq data. For custom code that has been deposited into a community repository, provide accession details.* |

# Flow Cytometry

## Plots

Confirm that:

☐ The axis labels state the marker and fluorochrome used (e.g. CD4-FITC).

☐ The axis scales are clearly visible. Include numbers along axes only for bottom left plot of group (a 'group' is an analysis of identical markers).

☐ All plots are contour plots with outliers or pseudocolor plots.

☐ A numerical value for number of cells or percentage (with statistics) is provided.

## Methodology

| | |
|---|---|
| Sample preparation | *Describe the sample preparation, detailing the biological source of the cells and any tissue processing steps used.* |
| Instrument | *Identify the instrument used for data collection, specifying make and model number.* |
| Software | *Describe the software used to collect and analyze the flow cytometry data. For custom code that has been deposited into a community repository, provide accession details.* |
| Cell population abundance | *Describe the abundance of the relevant cell populations within post-sort fractions, providing details on the purity of the samples and how it was determined.* |
| Gating strategy | *Describe the gating strategy used for all relevant experiments, specifying the preliminary FSC/SSC gates of the starting cell population, indicating where boundaries between "positive" and "negative" staining cell populations are defined.* |

☐ Tick this box to confirm that a figure exemplifying the gating strategy is provided in the Supplementary Information.

# Magnetic resonance imaging

## Experimental design

| | |
|---|---|
| Design type | *Indicate task or resting state; event-related or block design.* |
| Design specifications | *Specify the number of blocks, trials or experimental units per session and/or subject, and specify the length of each trial or block (if trials are blocked) and interval between trials.* |
| Behavioral performance measures | *State number and/or type of variables recorded (e.g. correct button press, response time) and what statistics were used to establish that the subjects were performing the task as expected (e.g. mean, range, and/or standard deviation across subjects).* |

## Acquisition

| | |
|---|---|
| Imaging type(s) | *Specify: functional, structural, diffusion, perfusion.* |
| Field strength | *Specify in Tesla* |
| Sequence & imaging parameters | *Specify the pulse sequence type (gradient echo, spin echo, etc.), imaging type (EPI, spiral, etc.), field of view, matrix size, slice thickness, orientation and TE/TR/flip angle.* |
| Area of acquisition | *State whether a whole brain scan was used OR define the area of acquisition, describing how the region was determined.* |

Diffusion MRI    ☐ Used    ☐ Not used

## Preprocessing

| | |
|---|---|
| Preprocessing software | *Provide detail on software version and revision number and on specific parameters (model/functions, brain extraction, segmentation, smoothing kernel size, etc.).* |
| Normalization | *If data were normalized/standardized, describe the approach(es): specify linear or non-linear and define image types used for transformation OR indicate that data were not normalized and explain rationale for lack of normalization.* |
| Normalization template | *Describe the template used for normalization/transformation, specifying subject space or group standardized space (e.g. original Talairach, MNI305, ICBM152) OR indicate that the data were not normalized.* |
| Noise and artifact removal | *Describe your procedure(s) for artifact and structured noise removal, specifying motion parameters, tissue signals and physiological signals (heart rate, respiration).* |

| Volume censoring | *Define your software and/or method and criteria for volume censoring, and state the extent of such censoring.* |
|---|---|

## Statistical modeling & inference

| Model type and settings | *Specify type (mass univariate, multivariate, RSA, predictive, etc.) and describe essential details of the model at the first and second levels (e.g. fixed, random or mixed effects; drift or auto-correlation).* |
|---|---|
| Effect(s) tested | *Define precise effect in terms of the task or stimulus conditions instead of psychological concepts and indicate whether ANOVA or factorial designs were used.* |

Specify type of analysis:  ☐ Whole brain  ☐ ROI-based  ☐ Both

| Statistic type for inference | *Specify voxel-wise or cluster-wise and report all relevant parameters for cluster-wise methods.* |
|---|---|

(See Eklund et al. 2016)

| Correction | *Describe the type of correction and how it is obtained for multiple comparisons (e.g. FWE, FDR, permutation or Monte Carlo).* |
|---|---|

## Models & analysis

| n/a | Involved in the study |
|---|---|
| ☐ | ☐ Functional and/or effective connectivity |
| ☐ | ☐ Graph analysis |
| ☐ | ☐ Multivariate modeling or predictive analysis |

| Functional and/or effective connectivity | *Report the measures of dependence used and the model details (e.g. Pearson correlation, partial correlation, mutual information).* |
|---|---|
| Graph analysis | *Report the dependent variable and connectivity measure, specifying weighted graph or binarized graph, subject- or group-level, and the global and/or node summaries used (e.g. clustering coefficient, efficiency, etc.).* |
| Multivariate modeling and predictive analysis | *Specify independent variables, features extraction and dimension reduction, model, training and evaluation metrics.* |

