## [Peer Review File · Nature Structural & Molecular Biology]

Peer Review Information

Manuscript Title: Structural and mechanistic insights into *Streptococcus pneumoniae* NADPH oxidase

Corresponding author name(s): Bonnie Murphy, Pablo San Segundo-Acosta

Reviewer Comments & Decisions:

Decision Letter, initial version:

Message: 4th Dec 2023

Dear Dr. Murphy,

Thank you again for submitting your manuscript "Structural and mechanistic insights into *Streptococcus pneumoniae* NADPH oxidase". We now have comments (below) from the 3 reviewers who evaluated your paper. In light of those reports, we remain interested in your study and would like to see your response to the comments of the referees, in the form of a revised manuscript.

You will see that while all three reviewers appreciate the work, reviewer #1 requests clarification on why the F397-in unproductive conformation of spNOX was exclusively captured, MD simulations to explore the energetics of the -in to -out productive state, and mutagenesis of residue Y136 to test its possible involvement in electron relay between the two hemes. Reviewer #2 asks whether the activity of the protein as an iron reductase can be verified, and Reviewer #3 asks for further clarification of the hydride transfer mechanism, and adding details to the methods on modeling using AlphaFold, and comparison of the results presented here with a competing structure of spNOX on BiorXiv. Please be sure to address/respond to all concerns of the referees in full in a point-by-point response and highlight all changes in the revised manuscript text file. If you have comments that are intended for editors only, please include those in a separate cover letter.

We expect to see your revised manuscript within 10-12 weeks. If you cannot send it

within this time, please contact us to discuss an extension; we would still consider your revision, provided that no similar work has been accepted for publication at NSMB or published elsewhere.

Reporting Summary:

When submitting the revised version of your manuscript, please pay close attention to our [href="https://www.nature.com/nature-portfolio/editorial-policies/image-integrity">Digital Image Integrity Guidelines](https://www.nature.com/nature-portfolio/editorial-policies/image-integrity). and to the following points below:

Please note that all key data shown in the main figures as cropped gels or blots should be presented in uncropped form, with molecular weight markers. These data can be aggregated into a single supplementary figure item. While these data can be displayed in a relatively informal style, they must refer back to the relevant figures. These data should be submitted with the final revision, as source data, prior to acceptance, but you may want to start putting it together at this point.

SOURCE DATA: we request that authors provide, in tabular form, the data underlying the graphical representations used in figures. This is to further increase transparency in data reporting, as detailed in this editorial (<http://www.nature.com/nsmb/journal/v22/n10/full/nsmb.3110.html>). Spreadsheets can be submitted in excel format. Only one (1) file per figure is permitted; thus, for multi-paneled figures, the source data for each panel should be clearly labeled in the Excel file; alternately the data can be provided as

multiple, clearly labeled sheets in an Excel file. When submitting files, the title field should indicate which figure the source data pertains to. We encourage our authors to provide source data at the revision stage, so that they are part of the peer-review process.

Data availability: this journal strongly supports public availability of data. All data used in accepted papers should be available via a public data repository, or alternatively, as Supplementary Information. If data can only be shared on request, please explain why in your Data Availability Statement, and also in the correspondence with your editor. Please note that for some data types, deposition in a public repository is mandatory - more information on our data deposition policies and available repositories can be found below: <https://www.nature.com/nature-research/editorial-policies/reporting-standards#availability-of-data>

[Redacted]

Sincerely,
Sara

Sara Osman, Ph.D.
Associate Editor
Nature Structural & Molecular Biology

Referee expertise:

Referee #1: Redox signalling, Cryo-EM

Referee #2: Oxidases, structural biology

Referee #3: Redox signalling, structural biology

Reviewers' Comments:

Reviewer #1:
Remarks to the Author:

NOX proteins play important physiological roles, however, their catalytic mechanism is not fully understood. The current work by Dubach et al. determined a series cryo-EM structure of bacterial NOX homolog spNOX up to 2.2 Å resolution, which is exceptionally high for 46 kDa membrane proteins. The structure shows a stable docking of DH at the bottom of TMD, it also shows how cofactors and substrates bind to spNOX. The work is a good complement to the previously published work on csNOX5, DUOX1, and NOX2. I have a few major concerns and minor concerns that might be helpful for the authors to improve their manuscript.

Major issues:

1. The cryo-EM sample of spNOX is apparently active in the enzymatic assay, but why the authors only captured the F397-in unproductive conformation? I would expect there is at least a fraction of particles showing the F397-out productive conformation.
2. It seems like the F397-binding mode is quite stable. So how can F397 move out of this low-energy state? An MD-simulation calculation might be necessary to answer this question.
3. Comparing the structure of wt spNOX and F397A mutant, there is little structural change of spNOX protein, except the conformation of the nicotinamide group of NADPH substrate. Would this reflect what happens during the catalysis of wt spNOX?
4. The sequence alignment around F397 in sFig. 8 is misleading to some extent. From Sfig.8, it seems that F397 is at the equivalent position to F1544 in DUOX1. But F1544 is on a stable beta structure in DUOX1. F397 is more like the F1551 at the C-terminus of DUOX1. I would suggest using more advanced sequence alignment tools, such as PROMALS3D for this tricky alignment.
5. In line 270, the authors conclude aromatic residues are not necessary for electron transfer between two hemes because the F107L mutation did not affect the activity of spNOX. Is it possible that in the F107L mutant, the conformation of Y136 at the same layer is somehow changed to mediate electron relay between two hemes? The authors might want to check F107 and Y136 double mutant.
6. Between FAD and inner heme, there are a few aromatic residues? Do they participate in the electron transfer between FAD and inner heme?

Minor issues:

1. The outer heme of spNOX is in a different pose compared to other NOX members. The author might elaborate more on this.
2. CBC atom of outer heme (507) has an alternative fitting which might be better than the current one.

Reviewer #2:

Remarks to the Author:

Dubach, Acosta, and Murphy present a comprehensive analysis of the cryoEM structure of a bacterial NADPH oxidase (NOX), shedding light on its mechanism. NOXs are membrane-bound enzymes producing reactive oxygen species (ROS) and utilize intracellular NAD(P)H to reduce oxygen, generating superoxide or H₂O₂ outside the cell. The authors exhibit exemplary experimental prowess, achieving high-resolution maps for a protein of less than 50 kDa—an experimental feat that underscores their expertise.

From a biochemical perspective, a pivotal finding emerges regarding NADPH binding. Based on existing literature, the authors engineer a mutant, facilitating the elucidation of the enzyme structure bound to nicotinamide in proximity to flavin. Consequently, the study establishes that NOXs operate through hydride transfer, debunking previous suggestions and aligning with the extensive knowledge based on ferredoxin reductases.

The structural analysis confirms the NOX structural conservation in dehydrogenase and trans-membrane domains, along with well-preserved FAD- and heme-binding sites. Notably, the oxygen-reacting center exhibits divergence from counterparts in other bacterial and human NOXs. The authors speculate that their protein may not be a conventional NOX but could function in the reduction of another protein. It could potentially serve as a ferric ion reductase—a notion left unexplored in the text and warrants experimental validation. Can the authors experimentally verify the protein activity as an iron reductase? Does the genomic context suggest the function?

In essence, this work contributes to our understanding of NOXs, particularly by clarifying the flavin reduction mechanism and affirming known structural features. However, the true substrate of the enzyme remains elusive. While endorsing the publication in journals such as Nature Communications, the manuscript may lack the depth and groundbreaking innovation typically expected for a Nature Structural Biology publication.

Reviewer #3:

Remarks to the Author:

A. Summary of the key results

Too generic and I am missing the importance of the study for the general NOX and Redox community. There is room for improvement.

B. Originality and significance: if not novel, please include reference

There is a similar PsNOX study published in BioRxiv:
<https://www.biorxiv.org/content/10.1101/2023.10.16.562591v1>

C. Data & methodology: validity of approach, quality of data, quality of presentation

Good, except for the kinetic data.

D. Appropriate use of statistics and treatment of uncertainties

Kinetic data should use SD iso SEM.

E. Conclusions: robustness, validity, reliability

See my text.

F. Suggested improvements: experiments, data for possible revision

See my text.

G. References: appropriate credit to previous work?

OK except for not mentioning the study of Petit-Härtlein et al.

H. Clarity and context: lucidity of abstract/summary, appropriateness of abstract, introduction and conclusions

See more details here below.

Dubash et al.'s manuscript marks a significant milestone in unraveling the substrate-bound structures of SpNOX, providing crucial insights into its catalytic mechanism. The study underscores a pivotal transient displacement of a C-terminal aromatic residue, a key element for hydride transfer—a phenomenon shared among members of the FNR superfamily. Noteworthy is the observation that the SpNOX Phe397Ala mutant adopts a productive conformation, correlating with enhanced turnover rates. Intriguingly, a parallel catalytic role has been suggested for a strictly conserved Phe residue at the C-terminus of eukaryotic NOXs, yet so far no published structures have illustrated its interaction with FAD.

Technically, the manuscript showcases the determination of high-resolution cryo-EM structures for the relatively compact SpNOX (46 kDa), underscoring advancements in microscopy, detectors, and software that propel the boundaries of single-particle cryo-EM.

While the narrative is compelling, the manuscript falls short in delving into the intricate details of substrate-bound structures and lacks a robust comparison with other NOXs in the discussion. Addressing this gap in the next review round will undoubtedly enhance the manuscript's depth and impact.

Several specific points merit attention. While the manuscript is well-crafted, it neglects a recent SpNOX study by Petit-Härtlein et al., which utilized X-ray crystallography and was uploaded to BioRxiv 3 days before the work of Dubash et al. A comparative discussion of these akin results, despite not sharing the exact same interpretation, could substantially enhance the manuscript's relevance to the NOX and redox communities. I recommend facilitating communication between both research groups to foster collaboration and

potentially refine interpretations.

Furthermore, clarifications are needed on the inaccurate claim of revealing new details on catalytic activity and electron-transfer pathways, as these have been previously demonstrated in eukaryotic NOXs.

Including a schematic overview of the hydride transfer mechanism and explaining abbreviations like DH in the main text are essential for reader comprehension.

Graphical and data representation concerns include the need to adjust Fig. 1b's x-axis to 50 μM for better visibility of the fitting in the crucial area. Fig. 2e requires improved clarity in the legend, necessitating adjustments in terms of a title that conveys a clear message. The steady-state kinetics data presented in Figure S14 necessitate accurate fitting using Hill coefficient equations. Additionally, there is an issue with the correctness of the KM units that requires attention and correction.

Crucial details on obtaining the AlphaFold model are missing from the methods section, and mechanistic intricacies, especially regarding electron transfer and FAD accepting two hydrides while NAD(P) accepts only one, remain unclear and merit detailed discussion.

Finally, a comprehensive exploration of the oxygen reduction mechanism, including speculation on potential two oxygen binding sites and electron tunneling, is warranted for a more thorough understanding of the manuscript's implications. Addressing these points will undoubtedly elevate the manuscript's overall quality and impact of this interesting study which I would like to see published in this journal.

Author Rebuttal to Initial comments

You will see that while all three reviewers appreciate the work, reviewer #1 requests clarification on why the F397-in unproductive conformation of spNOX was exclusively captured, MD simulations to explore the energetics of the -in to -out productive state, and mutagenesis of residue Y136 to test its possible involvement in electron relay between the two hemes. Reviewer #2 asks whether the activity of the protein as an iron reductase can be verified, and Reviewer #3 asks for further clarification of the hydride transfer mechanism, and adding details to the methods on modeling using AlphaFold, and comparison of the results presented here with a competing structure of spNOX on BiorXiv. Please be sure to address/respond to all concerns of the referees in full in a point-by-point response and highlight all changes in the revised manuscript text file. If you have comments that are intended for editors only, please include those in a separate cover letter.

We would really like to thank the editorial board very much for their work and for the realization of the interest of our study according to the comments of the reviewers. Added or moved text in the revised version of the manuscript has been highlighted in yellow. The revised version of the manuscript includes:

- A new panel in Fig. 2 (panel 2e) to show the interactions between the residues of the C-terminal tail of SpNOX and the TM domain.
- Changed Fig. 1b with a gap between the region of 100 and 250 μM (x-axis) for a better visibility of the fitting at lower substrate concentrations.
- Three new panels in Supplementary Fig. 13 (13d-f) comparing the NADPH-binding of human DUOX1 and SpNOX in more detail. Old panel 13d is now panel 13g.
- An update of the F397A and NADPH bound SpNOX models to address specific comments of the reviewers. Table 1 and Supplementary Fig. 7 have been updated accordingly.
- New figures in Supplementary Fig. 9 to address the differences between the outer heme of eukaryotic NOXs and SpNOX. Also new comments about this point in the revised version of the manuscript.
- A new figure (Supplementary Fig. 15) to show a schematic view of the proposed catalytic cycle for SpNOX, including the steps for hydride transfer and the displacement of Phe397. As a consequence, previous Supplementary Fig. 15 and 16 are now Supplementary Fig. 16 and 17.
- The kinetic characterization of two new mutants which were generated for the revised manuscript: a C-terminal deletion mutant ($\Delta 398-400$) and a F107L/Y136L double mutant, addressing specific reviewer questions. These mutants have been described in the manuscript.
- Updated Supplementary Tables 1 and 2 with the new oligonucleotides used to generate the new mutants.
- The ferric-reductase-activity assays of SpNOX (new Supplementary Figure 1 panels 1h and 1i).
- New error bars in the figures, which now represent the standard deviation (SD) instead of the standard error of the mean (SEM) as suggested by one of the reviewers.
- New improved alignments generated with the more advanced alignment tool PROMALS3D. Therefore, Supplementary Fig. 8a-c, Supplementary Fig.11 and the sequence identity and similarity values of the manuscript have been updated accordingly.
- New relevant residues for the manuscript discussions (i.e. the residues responsible for the interaction with the propionate groups of the outer hemes, or Lys250 for NADPH-interaction) have been highlighted in the newly generated alignments of Supplementary Figs. 8 and 11.
- We have added to the manuscript some comments and discussions addressing specific relevant findings of the preprint that describes the high-resolution crystal structure of the SpNOX DH domain and the low-resolution crystal structure of the inactive full-length F397W mutant (*Petit-Härtlein et al. X-Ray Structure and enzymatic study of a Bacterial NADPH oxidase highlight the activation mechanism of eukaryotic NOX. BioRxiv. 2023*).
- An analysis of the location of SpNOX into the *S. pneumoniae* genome was also performed, but it did not suggest any conclusive relation to the function of the protein, so it was not added to the revised version of the manuscript.
- The sentence " This conservation of FAD structure in NOXs, in spite of a change in the FAD-interacting amino acids, may point to an important role of this compact flavin conformation, in contrast to the commonly seen extended conformation in 286 other flavoproteins (e.g.

PDB:1B2R, 6HCY, 6LUM)” was deleted due to word number limitations and because we considered it was adding relevant value to the text.

Reviewer #1:

Remarks to the Author:

NOX proteins play important physiological roles; however, their catalytic mechanism is not fully understood. The current work by Dubach et al. determined a series cryo-EM structure of bacterial NOX homolog spNOX up to 2.2 Å resolution, which is exceptionally high for 46 kDa membrane proteins. The structure shows a stable docking of DH at the bottom of TMD, it also shows how cofactors and substrates bind to spNOX. The work is a good complement to the previously published work on csNOX5, DUOX1, and NOX2. I have a few major concerns and minor concerns that might be helpful for the authors to improve their manuscript.

We would really like to thank the reviewer very much for his/her work and for the useful suggestions pointed out, which after being addressed as indicated below have greatly strengthened the revised version of the manuscript in comparison to the original version of the manuscript.

Major issues:

1. The cryo-EM sample of spNOX is apparently active in the enzymatic assay, but why the authors only captured the F397-in unproductive conformation? I would expect there is at least a fraction of particles showing the F397-out productive conformation.

We agree with the reviewer that the productive conformation must be present at some level under turnover conditions; however, very short-lived intermediates will be present at very low levels even under turnover conditions, and it is not always possible to obtain a structure of such states by classification. Several previous studies of the conformational landscapes of different proteins under turnover conditions exemplify this phenomenon (cited below). For example, in the study by Hofmann et al., an outward-facing conformation of the TmrAB ABC transporter could never be resolved by cryo-EM under turnover conditions due to its highly transient nature. Moreover, recent theoretical work has indicated that, within the standard freezing conditions currently used in cryo-EM, free-energy barriers that cause transitions between the different conformational populations of the proteins in the sample can be overcome, generating a global range of conformations in the frozen sample that is more homogeneous than in the one before cooling (see Bock et al.). These limitations may be overcome using faster cooling conditions that are currently being implemented with the development of new time-resolved cryo-EM methods (see Torino, S. et al).

A comment about the possibility of the productive state being highly transient has been added to the revised manuscript (lines 304-306).

- Hofmann, S. et al. *Conformation space of a heterodimeric ABC exporter under turnover conditions. Nature* 571, 580–583 (2019).
- Parey, K. et al. *Cryo-EM structure of respiratory complex I at work. eLife* 7, e39213 (2018).
- Guo, H. & Rubinstein, J. L. *Structure of ATP synthase under strain during catalysis. Nat. Commun.* 13, 2232 (2022).
- Bock, L.V., Grubmüller, H. *Effects of cryo-EM cooling on structural ensembles. Nat Commun* 13, 1709 (2022).
- Torino, S., Dhurandhar, M., Stroobants, A. et al. *Time-resolved cryo-EM using a combination of droplet microfluidics with on-demand jetting. Nat Methods* 20, 1400–1408 (2023).

2. It seems like the F397-binding mode is quite stable. So how can F397 move out of this low-energy state? An MD-simulation calculation might be necessary to answer this question.

We thank the reviewer for his/her suggestion. We agree that MD simulations of SpNOX would be highly interesting to understand this and other aspects of NOX function. However, discussions with our theoretical colleagues on the topic led to the conclusion that carrying out accurate and valid MD simulation analyses would take a minimum of several months, and would be beyond the scope of the current paper, which presents substantial advances in understanding SpNOX from a structural and biochemical perspective.

Importantly, theoretical studies exist on the mechanism of hydride transfer in the canonical member of the ferredoxin-NADP⁺-reductase (FNR)-superfamily, the *Anabaena* FNR (see below). These studies focus on the catalytic role of *Anabaena* FNR C-terminal Tyr303, which sits between the isoalloxazine ring of FAD and the nicotinamide ring of NADP⁺ in the available WT crystal structures, similar to SpNOX Phe397. The simulations confirm that upon substrate binding, the complex evolves from a reactant state to a catalytically competent transition state, and finally to a product state. During the process, Tyr303 gets partially displaced to allow the nicotinamide ring to get closer, but it still lies between the lower moieties of the isoalloxazine and nicotinamide rings, producing an important destabilization (electrostatic and π - π stacking) of the otherwise close-contact ionic pair FADH⁽⁻⁾:NADP⁽⁺⁾. This has a positive kinetic effect for hydride transfer and makes the physiological reaction possible. Regarding other residues in the active site, these theoretical studies have focused on the role of Ser80, Cys261, and Glu301 (equivalent to SpNOX Ser236, Cys370 and Glu395), which only show minor rearrangements.

We anticipate that a similar situation may occur for SpNOX Phe397 (see the answer to the next question for further discussion about this point). We have added a comment on the potential of MD simulations to get more information about the mechanism of Phe397 displacement in lines 389-393:

- *Theoretical Study of the Mechanism of the Hydride Transfer between Ferredoxin–NADP⁺ Reductase and NADP⁺: The Role of Tyr303. Journal of the American Chemical Society 2012 134 (50), 20544-20553. DOI: 10.1021/ja310331v*

- *Peregrina, J.R., Lans, I. & Medina, M. The transient catalytically competent coenzyme allocation into the active site of Anabaena ferredoxin NADP⁺-reductase. Eur Biophys J 41, 117–128 (2012). <https://doi.org/10.1007/s00249-011-0704-5>*

3. Comparing the structure of wt spNOX and F397A mutant, there is little structural change of spNOX protein, except the conformation of the nicotinamide group of NADPH substrate. Would this reflect what happens during the catalysis of wt spNOX?

Currently, there isn't any structure available of a WT FNR family member in a productive conformation. However, the reaction mechanism of *Anabaena* ferredoxin-NADP⁺ reductase has been extensively studied, with a collection of WT and mutant enzymes complexed with NADP⁺ available, and several theoretical studies with molecular dynamic simulations (see above) based on these structures. These simulations in *Anabaena* FNR reveal that major changes occur in the distances and orientations between the rings of Tyr303, NADP⁺ and FAD. However, other relevant residues like Ser80, Cys261 and Glu301 only show minor rearrangements that might contribute to shorten the distance between the coenzymes.

Contrary to FNR Tyr303Ser mutant, which has a higher affinity for NADP⁺ than the WT and can only catalyze hydride transfer from NADPH to FAD, our SpNOX Phe397Ala mutant (and also the Phe397Ser mutant described in Petit-Härtlein et al. BioRxiv) does not show any increase in the

apparent affinity for NADPH and is able to catalyze hydride transfer faster than the WT. Besides, apart from the major changes in the position of the nicotinamide ring of NADPH, SpNOX Phe397Ala shows a rearrangement in the sidechain of Glu395 (equivalent to FNR Glu301). Compared to the inactive NADPH-bound WT structure, Glu395 interacts with nicotinamide N7N via an H-bond, probably contributing to the shortening of the distance between the nicotinamide and the isoalloxazine rings.

Therefore, considering that Phe397Ala SpNOX is able to catalyze hydride transfer, and that MD simulations of WT *Anabaena* FNR only show major changes in the position of the isoalloxazine, nicotinamide and C-terminal hydroxyphenyl rings, but not any other major rearrangements, the structure of Phe397Ala SpNOX may well reflect what happens during catalysis in WT SpNOX. Nevertheless, as discussed above, an in-depth study combining MD simulations with time-resolved cryo-EM studies could help to get further details, but as we also indicated above, in our belief it is out of the scope of the manuscript.

4. The sequence alignment around F397 in sFig. 8 is misleading to some extent. From Sfig.8, it seems that F397 is at the equivalent position to F1544 in DUOX1. But F1544 is on a stable beta structure in DUOX1. F397 is more like the F1551 at the C-terminus of DUOX1. I would suggest using more advanced sequence alignment tools, such as PROMALS3D for this tricky alignment.

We thank the reviewer for the suggestion. We have repeated in the revised version of the manuscript all alignments with PROMALS3D (see revised Supplementary Figs. 8a-c, and Supplementary Fig. 11). Of note, the alignment to DUOX1 is now more consistent with the structural picture.

5. In line 270, the authors conclude aromatic residues are not necessary for electron transfer between two hemes because the F107L mutation did not affect the activity of spNOX. Is it possible that in the F107L mutant, the conformation of Y136 at the same layer is somehow changed to mediate electron relay between two hemes? The authors might want to check F107 and Y136 double mutant.

Electron transfer rates between redox centers can be calculated by the quantum version of Marcus electron transfer theory, which depends on the edge-to-edge distance between the redox centers, the nuclear reorganization energy upon electron transfer (λ), the driving force for electron transfer (redox potentials) and the atom-packaging density (ρ) of the protein medium (see Page C. et al. Nature. 1999). On the basis of the multiple structural studies of redox proteins available, it has been found that redox proteins usually show designs that maintain redox centers at distances of ~ 14 Å or less, with most distances being between 6 and 10 Å. This suggests that redox proteins have evolved to have a structure in which redox centers are separated by distances below 14 Å, without having to optimize other factors that may influence electron-transfer rates. When these distances are kept, electron transfer by tunneling reaches rates fast enough not to be a rate-limiting step for the overall reaction (see Page CC. et al. Curr. Opin. Chem. Biol. 2003 below).

For eukaryotic NADPH oxidases and SpNOX, distances between the cofactors mediating single-electron-transfer steps (i.e. FAD and hemes) are maintained in the 8-11 Å range.

These distances should be short enough for electron transfer not to be rate-limiting, independent of the protein sequence. The distance between hemes and FAD-inner heme in SpNOX and other NADPH oxidases is between 8 and 10 Å and meets this criterion. For hemes of similar redox potentials separated by a distance of 10 Å, and assuming values of λ and ρ consistent with the literature (Page et al. 1999), Marcus's theory predicts an electron transfer rate higher than 10^6 s⁻¹; many orders of magnitude larger than the measured turnover rates of SpNOX or any related catalyst. In this context, it is hard to imagine that conserved residues meaningfully affect SpNOX activity via an effect on inter-heme electron transfer rates.

This is in line with our result for the F107L mutant, for which we did not observe any important change in the catalytic properties. Besides, as suggested by this reviewer, a double mutant with F107L and Y136L has been generated and its steady-state kinetics calculated. The mutant shows a similar activity to WT and F107L, suggesting that neither F107 nor Y136 residues are crucial in the electron transfer between the two heme cofactors.

In this context, to address the concern of the reviewer, the Michaelis-Menten-fitted curve of the double mutant has been added in the revised version of Supplementary Fig. 14 (panel 14f). Therefore, we think that the aromatic residue conserved in NADPH oxidases is not an essential path for efficient electron transfer, and a reduction of protein activity in eukaryotic NADPH oxidases when this residue is mutated could reflect structural changes. Additionally, these results have been discussed in lines 286-297 of the revised version of the manuscript.

Page, C., Moser, C., Chen, X. et al. Natural engineering principles of electron tunnelling in biological oxidation–reduction. Nature 402, 47–52 (1999). <https://doi.org/10.1038/46972>

Page CC, Moser CC, Dutton PL. Mechanism for electron transfer within and between proteins. Curr Opin Chem Biol. 2003 Oct;7(5):551-6. doi: 10.1016/j.cbpa.2003.08.005. PMID: 14580557.

6. Between FAD and inner heme, there are a few aromatic residues? Do they participate in the electron transfer between FAD and inner heme?

As stated in the previous answer, there is no need for SpNOX to have specific residues mediating an efficient electron transfer between FAD and the inner heme. Besides, these residues are part of the FAD-binding lobe (e.g. Tyr122), or seem to be mediating the docking site between the dehydrogenase (DH) domain and the transmembrane (TM) domain (e.g. Phe399). Therefore, we expect that mutations in these residues lead to alterations in FAD binding (FAD structure, FAD-binding affinity or even FAD-redox potential), and/or in DH and TM-domain docking. This would affect parameters related to electron tunneling rates like edge-to-edge distances, driving force, nuclear reorganization or packing density, but would not mean that these residues are a preferred nor essential path for these electrons.

In this context, to address the concern of the reviewer, we have produced a C-terminal-deletion SpNOX mutant (Δ 398-400) and analyzed its steady-state kinetics with NADPH as substrate (see Supplementary Fig. 14a). We observe a modest increase in K_M and a modest decrease in k_{cat} . This leads to a reduction of the catalytic efficiency to half with respect to the WT. As both parameters seem to be affected, we think that the removal of the tail may be affecting the docking of the DH and TM domains, and NADPH-binding as a consequence. These results have been included in the revised version of the manuscript (lines 236-240) and as a new panel in Fig 2 (panel e). We have not generated mutants of Tyr122 or Trp125 because we think they are involved in FAD-binding and most probably in the folding of the TM domain too, so the results (likely a reduction in apparent k_{cat}) would not be conclusive. In fact, a Tyr122Ala mutant has been characterized in the SpNOX study of Petit-Härtlein et al, and shows a six-fold reduction in the affinity for FAD and a reduction in k_{cat} . Thus, a comment regarding this mutant has been additionally included in the revised version of the manuscript (lines 226-228). Besides, an Arg126Ala mutant was characterized in the Petit-Härtlein et al study. This mutant shows a strongly decreased affinity for FAD, in spite of interacting directly only with the inner heme in our high-resolution structures. This suggests that the amino acids involved in FAD and inner-heme binding are relevant to preserve the whole DH and TM domain interface, and therefore mutations in these amino acids would compromise the structure and catalytic activity of SpNOX.

Petit-Härtlein et al. X-Ray Structure and enzymatic study of a Bacterial NADPH oxidase highlight the activation mechanism of eukaryotic NOX. BioRxiv. 2023.

Minor issues:

1. The outer heme of spNOX is in a different pose compared to other NOX members. The author might elaborate more on this.

We thank the reviewer for this comment. To address the concern of the reviewer, we have updated Supplementary Fig. 9 with a comparison of the outer hemes. Additionally, we added a comment about this difference in the Results section, where the TM domain of SpNOX is compared to the one of eukaryotic NOXs (see lines 202-205). Besides, we have highlighted the residues mediating the interaction with the propionate groups of the outer heme in the revised version of Supplementary Fig. 8a.

2. CBC atom of outer heme (507) has an alternative fitting which might be better than the current one.

We thank the reviewer for this comment. We have checked and updated the models according to the suggestion of the reviewer.

Reviewer #2:

Remarks to the Author:

Dubach, Acosta, and Murphy present a comprehensive analysis of the cryoEM structure of a bacterial NADPH oxidase (NOX), shedding light on its mechanism. NOXs are membrane-bound enzymes producing reactive oxygen species (ROS) and utilize intracellular NAD(P)H to reduce oxygen, generating superoxide or H₂O₂ outside the cell. The authors exhibit exemplary experimental prowess, achieving high-resolution maps for a protein of less than 50 kDa—an experimental feat that underscores their expertise.

From a biochemical perspective, a pivotal finding emerges regarding NADPH binding. Based on existing literature, the authors engineer a mutant, facilitating the elucidation of the enzyme structure bound to nicotinamide in proximity to flavin. Consequently, the study establishes that NOXs operate through hydride transfer, debunking previous suggestions and aligning with the extensive knowledge based on ferredoxin reductases.

The structural analysis confirms the NOX structural conservation in dehydrogenase and trans-membrane domains, along with well-preserved FAD- and heme-binding sites. Notably, the oxygen-reacting center exhibits divergence from counterparts in other bacterial and human NOXs. The authors speculate that their protein may not be a conventional NOX but could function in the reduction of another protein. It could potentially serve as a ferric ion reductase—a notion left unexplored in the text and warrants experimental validation. Can the authors experimentally verify the protein activity as an iron reductase? Does the genomic context suggest the function?

In essence, this work contributes to our understanding of NOXs, particularly by clarifying the flavin reduction mechanism and affirming known structural features. However, the true substrate of the enzyme remains elusive. While endorsing the publication in journals such as Nature Communications, the manuscript may lack the depth and groundbreaking innovation typically expected for a Nature Structural Biology publication.

We thank the reviewer for his/her work, and these useful comments that have been addressed in the revised version of the manuscript.

In the original study describing SpNOX, ferric reductase activity was examined and found to be comparable to the ferric reductase activity of human NOX2 (Hajjar *et al.* 2017). This assay was performed under aerobic conditions and in the presence of superoxide dismutase (SOD) to minimize the reduction of Fe(III) by the produced superoxide. The amount of Fe(III) reduction with SOD present was minimal, and the idea of SpNOX being a true ferric reductase was discarded. However, in this paper we show that SpNOX can directly transfer electrons to SOD, invalidating the previous activity assay results carried out in its presence, as SOD would also act as a competitor to the ferrous iron at high concentrations, in addition to scavenging any superoxide formed. Moreover, in the revised version of the manuscript, we show SpNOX does exhibit direct electron transfer to ferrous iron by performing the assay under anaerobic conditions (added as Supplementary Fig. 1h). However, the reduction rate (3.6 min⁻¹) is 40-times slower than for cytochrome c (124.7 min⁻¹) or 10-times slower than atmospheric oxygen (48 min⁻¹, see lines 163-164). We further performed Michaelis-Menten kinetics analysis for the Fe(III) reduction (added as Supplementary Fig 1i), showing a low apparent k_{cat} (0.05 s⁻¹) and a high apparent K_M (81.3 μM) which result in a very low catalytic efficiency (k_{cat}/K_M , 0.00063 s⁻¹ μM⁻¹) (see lines 165-169). Altogether, these results speak strongly against Fe(III) as the physiological electron acceptor.

We have carried out an analysis of the genomic context of the SpNOX gene (SPNHU17_RS03015, see attached image) in one of the available *S. pneumoniae* reference genomes at NCBI (NZ_CP020549.1). The SpNOX gene is located near a vancomycin resistance operon (*vncR*, *vncS*), the glycolytic fructose bis-phosphate aldolase gene (SPNHU17_RS03010) and an operon

of ABC transporters and permeases potentially involved in glutamine transport (SPNHU17_RS03020, RS03025 and others). Although we cannot rule out a meaningful connection to the function of any of these genes, a general connection to known or putative reactivity of SpNOX is not apparent.

The position of SpNOX gene SPNHU17_RS03015 in the *S. pneumoniae* genome

Hajjar, C., et al., *The NOX Family of Proteins Is Also Present in Bacteria*. mBio, 2017. **8**(6): p. e01487-17.

A link to the genomic location of SpNOX:

https://www.ncbi.nlm.nih.gov/gene?cmd=retrieve&list_uids=66805763

Reviewer #3:

Remarks to the Author:

A. Summary of the key results

We thank the reviewer for his/her work, the realization of the interest of the work, and his/her suggestions to improve the presented work. Moreover, as suggested, we have updated the Abstract to give more specific information about the findings of our study and their relevance to the redox community.

B. Originality and significance: if not novel, please include reference

There is a similar PsNOX study published in

BioRxiv: <https://www.biorxiv.org/content/10.1101/2023.10.16.562591v1>

C. Data & methodology: validity of approach, quality of data, quality of presentation

Good, except for the kinetic data.

D. Appropriate use of statistics and treatment of uncertainties

Kinetic data should use SD iso SEM.

We have made these changes, thank you.

E. Conclusions: robustness, validity, reliability

See my text.

F. Suggested improvements: experiments, data for possible revision

See my text.

G. References: appropriate credit to previous work?

OK except for not mentioning the study of Petit-Härtlein et al.

We thank the reviewer for this point. The study of Petit-Härtlein et al. was uploaded to BioRxiv after we submitted the manuscript to NSMB, so that it was not possible for us to include a comparison to this study in our original manuscript. We have now referred to this study in the following parts of our revised version of the manuscript, as we think it adds relevance to our results:

- The existence of a manuscript describing a high-resolution structure of the DH domain of SpNOX and a low-resolution structure of full-length SpNOX Phe397Trp is mentioned in lines 182-185 of the revised manuscript.
- The high-affinity of SpNOX for FAD, and its lower affinity for smaller, less bulky flavins like flavin mononucleotide (FMN), is mentioned in lines 217-218, where we describe the FAD-binding site.
- The Tyr122Ala mutant characterized in that study is referenced in lines 226-228 to confirm the relevance of Tyr122 in FAD binding.
- The discussion made in that study regarding the activation of eukaryotic NOXs to achieve hydride transfer by tensioning of the NADPH-binding lobe is mentioned in lines 274-282, and supported by the new panels 13d-f of Supplementary Figure 13 comparing NADPH-binding in SpNOX and high-calcium human DUOX1.

- In the Petit-Härtlein et al. study, two Phe397 mutants were characterized: a Phe397Ser mutant with equivalent steady-state kinetics to our Phe397Ala mutant; and an inactive Phe397Trp mutant. We have mentioned these mutants in the Results (lines 182-185 and 323-325) and Discussion (lines 389-392) sections, where we describe the potential hydride transfer mechanism of SpNOX.

It is important to note that the study by Petit-Härtlein et al. did not achieve comparable resolution values to ours. This fact has been a limiting factor and has led to incorrect interpretations about the role of some residues, including Arg126 and Tyr136. Other differences, like a further distance between Phe397 and the isoalloxazine ring in their high-resolution structure of the DH domain, could be explained as a consequence of the missing TM domain. This domain is involved in FAD binding by Tyr122, and in positioning the C-terminal tail orthogonally to the membrane. Additionally, the adenine of the FAD has been misplaced in the full-length structure due to taking the FAD geometry from the high-resolution DH domain structure, which lacks the crucial residues for correct binding (i.e. Tyr122). Due to the fact that the work of Petit-Härtlein et al. does not include any full-length nicotinamide-bound structure, so that a detailed comparison is not possible.

H. Clarity and context: lucidity of abstract/summary, appropriateness of abstract, introduction and conclusions

See more details here below.

Dubash et al.'s manuscript marks a significant milestone in unraveling the substrate-bound structures of SpNOX, providing crucial insights into its catalytic mechanism. The study underscores a pivotal transient displacement of a C-terminal aromatic residue, a key element for hydride transfer—a phenomenon shared among members of the FNR superfamily. Noteworthy is the observation that the SpNOX Phe397Ala mutant adopts a productive conformation, correlating with enhanced turnover rates. Intriguingly, a parallel catalytic role has been suggested for a strictly conserved Phe residue at the C-terminus of eukaryotic NOXs, yet so far no published structures have illustrated its interaction with FAD.

Technically, the manuscript showcases the determination of high-resolution cryo-EM structures for the relatively compact SpNOX (46 kDa), underscoring advancements in microscopy, detectors, and software that propel the boundaries of single-particle cryo-EM.

While the narrative is compelling, the manuscript falls short in delving into the intricate details of substrate-bound structures and lacks a robust comparison with other NOXs in the discussion. Addressing this gap in the next review round will undoubtedly enhance the manuscript's depth and impact.

We thank the reviewer for this comment. We have revised the manuscript according to the suggestions of the reviewer, and now the following figures and parts of the text compare SpNOX with other NOXs, giving details about specific structural aspects:

→In the "*Architecture of SpNOX*" section of the *Results*:

- A general comparison of the global domain arrangement and sequences of SpNOX and the other NOXs is described in lines 189-193, and supported by Fig. 1 and Supplementary Fig. 8a-c.
- A comparison between the binding to the outer heme between SpNOX, eukaryotic NOXs and csNOX5 has been added to lines 202-205, and Supplementary Figs. 8a and 9.
- A detailed comparison between the whole DH domains and the FAD-binding regions is made in Supplementary Fig. 8a-c, 9, 10d-f and in lines 206-216 and 217-233.

→ In the section “*The NAD(P)H-binding site and the electron transfer pathway of SpNOX*” of the *Results*:

- A detailed description of the NAD(P)H-binding site of SpNOX is given in lines 244-253 and lines 254-282.
- To our knowledge, the only published study of a NOX in a substrate-bound configuration is for human DUOX1. There are also structures of other members of the FNR superfamily with NADPH bound. In lines 259-270, we compare the NAD(P)H-binding site of SpNOX to the sites of high-calcium human DUOX1, *Anabaena* FNR, and the predicted site of NOX2, to explain in detail the lack of NADPH specificity in SpNOX. These descriptions and comparisons are supported by Figures 3a-b, Supplementary Fig. 8a-c and Supplementary Fig. 9.
- We have now added a more detailed comparison between the NADPH-binding sites of human DUOX1 and SpNOX, and added a reference to the Petit-Härtlein et al. preprint (lines 274-282 and Supplementary Fig. 13, including new panels 13d-f).
- A comparison between the hydrophobic cluster present between hemes in SpNOX and the other NOXs, is made in lines 286-300.
- The search of potential oxygen-binding sites is described in lines 334-352.

→ In the Discussion, we made comparisons between SpNOX and eukaryotic NOXs focused on the role of the C-terminal aromatic residue (lines 384-404), and on the potential oxygen reduction centers (lines 410-422). These discussions are supported by Figs. 3 and 4, and Supplementary Figs. 9 and 10.

Several specific points merit attention. While the manuscript is well-crafted, it neglects a recent SpNOX study by Petit-Härtlein et al., which utilized X-ray crystallography and was uploaded to BioRxiv 3 days before the work of Dubash et al. A comparative discussion of these akin results, despite not sharing the exact same interpretation, could substantially enhance the manuscript's relevance to the NOX and redox communities. I recommend facilitating communication between both research groups to foster collaboration and potentially refine interpretations.

Please see our response above.

Furthermore, clarifications are needed on the inaccurate claim of revealing new details on catalytic activity and electron-transfer pathways, as these have been previously demonstrated in eukaryotic NOXs.

In the text, we already mention that the proposed electron-transfer pathway in SpNOX is similar to the previously described in other NOXs (lines 283-285). However, we do think that we have also made several new contributions that add relevant information not only for the electron-transfer pathway, but also for the catalytic mechanism. Here, we highlight the following specific points, addressed in the manuscript, which inform our understanding of catalytic activity and electron transfer within NADPH oxidases:

- Structural and mutational evidence for the role of Phe397 in the reaction. These experiments confirm that NOX enzymes use a hydride transfer mechanism for electron transfer from NAD(P)H to FAD, like other members of the FNR superfamily (lines 301-333).
- Analysis of the potential role of the C-terminal tail residues in the docking of the DH and TM domains (added to the revised version of the manuscript, lines 236-240).
- Mutational analysis of the (absence of) role of Phe107 or Tyr136 in efficient electron tunneling between heme groups, in agreement with the quantum version of Marcus electron transfer theory (lines 292-300).

-Structural and mutational analysis of substrate specificity, which gives residue-level insights into the mechanism of NADPH and NADH binding in SpNOX (lines 259-270). Three more technical replicates (same protein batch) were added to the kinetic characterization of the Y353R SpNOX mutant to get a better fitted curve and more precise kinetic parameters (Supplementary Fig. 14c).

Including a schematic overview of the hydride transfer mechanism

To address the concern of the reviewer, we have included that schematic overview as Supplementary Fig. 15 and mentioned it in the Discussion section of the revised version of the manuscript (line 392-393).

and explaining abbreviations like DH in the main text are essential for reader comprehension.

We have carefully checked that we define all abbreviations when they are first used.

Graphical and data representation concerns include the need to adjust Fig. 1b's x-axis to 50 μ M for better visibility of the fitting in the crucial area. Fig. 2e requires improved clarity in the legend, necessitating adjustments in terms of a title that conveys a clear message.

To address this issue, we have adjusted the x-axis in Fig. 1b with a gap between the region of 100 and 250 μ M for a better visibility of the fitting at lower substrate concentrations.

The steady-state kinetics data presented in Figure S14 necessitate accurate fitting using Hill coefficient equations.

Given that spNOX is a monomeric protein, and given that our structural analysis in the presence of NADPH revealed a single NADPH-binding site, it is difficult to imagine that there are any relevant cooperativity effects. For this reason, we think a Hill coefficient of 1 is appropriate in this context.

Additionally, there is an issue with the correctness of the KM units that requires attention and correction.

Fixed in the revised version of the manuscript. Thank you.

Crucial details on obtaining the AlphaFold model are missing from the methods section,

The AlphaFold2 models shown in this work were downloaded from the AlphaFold Protein Structure Database created by DeepMind in collaboration with the EMBL.

To address this issue, we have cited this paper, where specific details for the generation of the models can be found, in the methods section (lines 643-644).

Varadi M., et al. AlphaFold Protein Structure Database: massively expanding the structural coverage of protein-sequence space with high-accuracy models, Nucleic Acids Research, Volume 50, Issue D1, 7 January 2022, Pages D439–D444

and mechanistic intricacies, especially regarding electron transfer and FAD accepting two hydrides while NAD(P) accepts only one, remain unclear and merit detailed discussion.

We are not aware of precedent for the idea that FAD can accept two hydrides, and a careful analysis of the scientific literature did not reveal any studies suggesting that it can. Perhaps the reviewer meant to say 'FAD accepts two electrons'?, in which case we agree; in particular, the

precedent established for redox reactions involving nicotinamide as substrate is that a hydride (two electrons and one proton) is transferred to the flavin, which can undergo two-electron (hydride) and one-electron transitions.

In this context, to address this issue, we have added in the revised version of the manuscript a scheme of the proposed catalytic cycle of SpNOX in Supplementary Fig. 15.

Finally, a comprehensive exploration of the oxygen reduction mechanism, including speculation on potential two oxygen binding sites and electron tunneling, is warranted for a more thorough understanding of the manuscript's implications. Addressing these points will undoubtedly elevate the manuscript's overall quality and impact of this interesting study which I would like to see published in this journal.

We thank the reviewer for this comment. The two potential oxygen-binding sites, how they were analyzed, and the mutagenic analyses are included in the Results section of the manuscript (lines 334-352) and in the Discussion section (412-437). Although the results of these analyses do not allow us to draw confident conclusions on the nature or binding site of the final acceptor, we think that they point towards a role for SpNOX different from ROS production and more related to the reduction of other substrates, most probably bulky substrates like proteins. Regarding electron tunneling from FAD to the transmembrane hemes, we have produced a new double mutant (F107L/Y136L) for the revised version of the manuscript (lines 290-300) and new Supplementary Fig. 14f), and we have concluded that these aromatic residues between hemes are not a preferred route for electron tunneling in SpNOX. This is agreement with previous studies (see *Page C. et al* below) showing that distances below 14 Å between redox centers transferring single electrons are enough to reach electron-transfer rates (calculated by the quantum version of the Marcus theory of electron tunneling) several orders of magnitude faster than redox reactions.

Page, C., Moser, C., Chen, X. et al. Natural engineering principles of electron tunnelling in biological oxidation–reduction. Nature 402, 47–52 (1999). <https://doi.org/10.1038/46972>

Page CC, Moser CC, Dutton PL. Mechanism for electron transfer within and between proteins. Curr Opin Chem Biol. 2003 Oct;7(5):551-6. doi: 10.1016/j.cbpa.2003.08.005. PMID: 14580557.

Decision Letter, first revision:

Message: Our ref: NSMB-A48331A

22nd Feb 2024

Dear Dr. Murphy,

Thank you for submitting your revised manuscript "Structural and mechanistic insights into *Streptococcus pneumoniae* NADPH oxidase" (NSMB-A48331A). It has now been seen by the original referees and their comments are below. The reviewers find that the paper has improved in revision, and therefore we'll be happy in principle to publish it in Nature Structural & Molecular Biology, pending minor revisions to satisfy the referees' final requests and to comply with our editorial and formatting guidelines.

We are now performing detailed checks on your paper and will send you a checklist detailing our editorial and formatting requirements in the next few weeks. Please do not upload the final materials and make any revisions until you receive this additional information from us.

To facilitate our work at this stage, it is important that we have a copy of the main text as a word file. If you could please send along a word version of this file as soon as possible, we would greatly appreciate it; please make sure to copy the NSMB account (cc'ed above).

Sincerely,
Sara

Sara Osman, Ph.D.
Associate Editor
Nature Structural & Molecular Biology

Reviewer #1 (Remarks to the Author):

I agree that the MD simulation is not mandatory.
I have no more questions.

Reviewer #2 (Remarks to the Author):

The revisions made by the authors perfectly address the questions and issues raised by the reviewers. Consequently, the manuscript has been further improved and written with great clarity. This work will contribute to our understanding of NOXs and their mechanisms of function.

Reviewer #3 (Remarks to the Author):

Summary of Key Results:

The revised manuscript presents a comprehensive investigation in which Victor Dubach et al. is looking into the mechanistic insights of an NADPH oxidase using cryo-EM as main technique. The study elucidates the role of Phe397 in hydride transfer to FAD, shedding light on the mechanism constitutive activity by using mutagenesis and biochemical techniques. The authors provide compelling evidence supporting their conclusions, which significantly advance our understanding of how the NADPH oxidase of *Streptococcus pneumoniae* in vivo could function.

Originality and Significance:

The high-resolution cryo-EM structures of this NOX obtained in substrate-free conditions and under reducing conditions with NADH and under turnover conditions with NADPH presented in their study are novel and it uncovers new insights, thereby contributing substantially to the field of NOX research. Their findings provide us insights into a bacterial NOX protein.

Data & Methodology:

I am pleased with the now more in depth discussion, and data presented in this revised manuscript. Overall, the methodology utilized aligns well with the objectives of the study and appears to be valid. Particularly noteworthy is the clear presentation of the structural model figures, which greatly enhances comprehension.

Conclusions:

The conclusions drawn by the authors are robust and supported by the evidence presented. The conclusions are consistent with the study's objectives and contribute significantly to the advancement of knowledge in the field.

Suggested Improvements:

The manuscript demonstrates a high level of quality, requiring no further improvements. However, I would suggest that the authors incorporate Supplementary Figure 15 into the main manuscript. This figure illustrates the proposed catalytic cycle and the mechanism of hydride transfer, effectively summarizing the study in a compelling manner.

References:

The revised manuscript appropriately credits previous published work.

Clarity and Context:

The adapted abstract in the revised manuscript succinctly summarizes the key findings and objectives of the study, effectively capturing the essence of the research. The conclusions drawn are logical and well-supported by the data presented, effectively tying together the various components of the manuscript.

In conclusion, the manuscript submitted by Dr. Bonnie Murphy's lab represents a significant contribution to the NOX redox field. The originality, rigorous methodology, and compelling conclusions make it well-suited for publication in *Nature Structural & Molecular Biology*.

Author Rebuttal, first revision:

You will see that while all three reviewers appreciate the work, reviewer #1 requests clarification on why the F397-in unproductive conformation of spNOX was exclusively captured, MD simulations to explore the energetics of the -in to -out productive state, and mutagenesis of residue Y136 to test its possible involvement in electron relay between the two hemes. Reviewer #2 asks whether the activity of the protein as an iron reductase can be verified, and Reviewer #3 asks for further clarification of the hydride transfer mechanism, and adding details to the methods on modeling using AlphaFold, and comparison of the results presented here with a competing structure of spNOX on BiorXiv. Please be sure to address/respond to all concerns of the referees in full in a point-by-point response and highlight all changes in the revised manuscript text file. If you have comments that are intended for editors only, please include those in a separate cover letter.

We would really like to thank the editorial board very much for their work and for the realization of the interest of our study according to the comments of the reviewers. Added or moved text in the revised version of the manuscript has been highlighted in yellow. The revised version of the manuscript includes:

- A new panel in Fig. 2 (panel 2e) to show the interactions between the residues of the C-terminal tail of SpNOX and the TM domain.
- Changed Fig. 1b with a gap between the region of 100 and 250 μM (x-axis) for a better visibility of the fitting at lower substrate concentrations.
- Three new panels in Supplementary Fig. 13 (13d-f) comparing the NADPH-binding of human DUOX1 and SpNOX in more detail. Old panel 13d is now panel 13g.
- An update of the F397A and NADPH bound SpNOX models to address specific comments of the reviewers. Table 1 and Supplementary Fig. 7 have been updated accordingly.
- New figures in Supplementary Fig. 9 to address the differences between the outer heme of eukaryotic NOXs and SpNOX. Also new comments about this point in the revised version of the manuscript.
- A new figure (Supplementary Fig. 15) to show a schematic view of the proposed catalytic cycle for SpNOX, including the steps for hydride transfer and the displacement of Phe397. As a consequence, previous Supplementary Fig. 15 and 16 are now Supplementary Fig. 16 and 17.
- The kinetic characterization of two new mutants which were generated for the revised manuscript: a C-terminal deletion mutant ($\Delta 398-400$) and a F107L/Y136L double mutant, addressing specific reviewer questions. These mutants have been described in the manuscript.
- Updated Supplementary Tables 1 and 2 with the new oligonucleotides used to generate the new mutants.
- The ferric-reductase-activity assays of SpNOX (new Supplementary Figure 1 panels 1h and 1i).
- New error bars in the figures, which now represent the standard deviation (SD) instead of the standard error of the mean (SEM) as suggested by one of the reviewers.
- New improved alignments generated with the more advanced alignment tool PROMALS3D. Therefore, Supplementary Fig. 8a-c, Supplementary Fig.11 and the sequence identity and similarity values of the manuscript have been updated accordingly.
- New relevant residues for the manuscript discussions (i.e. the residues responsible for the interaction with the propionate groups of the outer hemes, or Lys250 for NADPH-interaction) have been highlighted in the newly generated alignments of Supplementary Figs. 8 and 11.
- We have added to the manuscript some comments and discussions addressing specific relevant findings of the preprint that describes the high-resolution crystal structure of the SpNOX DH domain and the low-resolution crystal structure of the inactive full-length F397W mutant (*Petit-Härtlein et al. X-Ray Structure and enzymatic study of a Bacterial NADPH oxidase highlight the activation mechanism of eukaryotic NOX. BioRxiv. 2023*).
- An analysis of the location of SpNOX into the *S. pneumoniae* genome was also performed, but it did not suggest any conclusive relation to the function of the protein, so it was not added to the revised version of the manuscript.
- The sentence " This conservation of FAD structure in NOXs, in spite of a change in the FAD-interacting amino acids, may point to an important role of this compact flavin conformation, in contrast to the commonly seen extended conformation in 286 other flavoproteins (e.g.

PDB:1B2R, 6HCY, 6LUM)” was deleted due to word number limitations and because we considered it was adding relevant value to the text.

Reviewer #1:

Remarks to the Author:

NOX proteins play important physiological roles; however, their catalytic mechanism is not fully understood. The current work by Dubach et al. determined a series cryo-EM structure of bacterial NOX homolog spNOX up to 2.2 Å resolution, which is exceptionally high for 46 kDa membrane proteins. The structure shows a stable docking of DH at the bottom of TMD, it also shows how cofactors and substrates bind to spNOX. The work is a good complement to the previously published work on csNOX5, DUOX1, and NOX2. I have a few major concerns and minor concerns that might be helpful for the authors to improve their manuscript.

We would really like to thank the reviewer very much for his/her work and for the useful suggestions pointed out, which after being addressed as indicated below have greatly strengthened the revised version of the manuscript in comparison to the original version of the manuscript.

Major issues:

1. The cryo-EM sample of spNOX is apparently active in the enzymatic assay, but why the authors only captured the F397-in unproductive conformation? I would expect there is at least a fraction of particles showing the F397-out productive conformation.

We agree with the reviewer that the productive conformation must be present at some level under turnover conditions; however, very short-lived intermediates will be present at very low levels even under turnover conditions, and it is not always possible to obtain a structure of such states by classification. Several previous studies of the conformational landscapes of different proteins under turnover conditions exemplify this phenomenon (cited below). For example, in the study by Hofmann et al., an outward-facing conformation of the TmrAB ABC transporter could never be resolved by cryo-EM under turnover conditions due to its highly transient nature. Moreover, recent theoretical work has indicated that, within the standard freezing conditions currently used in cryo-EM, free-energy barriers that cause transitions between the different conformational populations of the proteins in the sample can be overcome, generating a global range of conformations in the frozen sample that is more homogeneous than in the one before cooling (see Bock et al.). These limitations may be overcome using faster cooling conditions that are currently being implemented with the development of new time-resolved cryo-EM methods (see Torino, S. et al).

A comment about the possibility of the productive state being highly transient has been added to the revised manuscript (lines 304-306).

- Hofmann, S. et al. *Conformation space of a heterodimeric ABC exporter under turnover conditions. Nature* 571, 580–583 (2019).
- Parey, K. et al. *Cryo-EM structure of respiratory complex I at work. eLife* 7, e39213 (2018).
- Guo, H. & Rubinstein, J. L. *Structure of ATP synthase under strain during catalysis. Nat. Commun.* 13, 2232 (2022).
- Bock, L.V., Grubmüller, H. *Effects of cryo-EM cooling on structural ensembles. Nat Commun* 13, 1709 (2022).
- Torino, S., Dhurandhar, M., Stroobants, A. et al. *Time-resolved cryo-EM using a combination of droplet microfluidics with on-demand jetting. Nat Methods* 20, 1400–1408 (2023).

2. It seems like the F397-binding mode is quite stable. So how can F397 move out of this low-energy state? An MD-simulation calculation might be necessary to answer this question.

We thank the reviewer for his/her suggestion. We agree that MD simulations of SpNOX would be highly interesting to understand this and other aspects of NOX function. However, discussions with our theoretical colleagues on the topic led to the conclusion that carrying out accurate and valid MD simulation analyses would take a minimum of several months, and would be beyond the scope of the current paper, which presents substantial advances in understanding SpNOX from a structural and biochemical perspective.

Importantly, theoretical studies exist on the mechanism of hydride transfer in the canonical member of the ferredoxin-NADP⁺-reductase (FNR)-superfamily, the *Anabaena* FNR (see below). These studies focus on the catalytic role of *Anabaena* FNR C-terminal Tyr303, which sits between the isoalloxazine ring of FAD and the nicotinamide ring of NADP⁺ in the available WT crystal structures, similar to SpNOX Phe397. The simulations confirm that upon substrate binding, the complex evolves from a reactant state to a catalytically competent transition state, and finally to a product state. During the process, Tyr303 gets partially displaced to allow the nicotinamide ring to get closer, but it still lies between the lower moieties of the isoalloxazine and nicotinamide rings, producing an important destabilization (electrostatic and π - π stacking) of the otherwise close-contact ionic pair FADH⁽⁻⁾:NADP⁽⁺⁾. This has a positive kinetic effect for hydride transfer and makes the physiological reaction possible. Regarding other residues in the active site, these theoretical studies have focused on the role of Ser80, Cys261, and Glu301 (equivalent to SpNOX Ser236, Cys370 and Glu395), which only show minor rearrangements.

We anticipate that a similar situation may occur for SpNOX Phe397 (see the answer to the next question for further discussion about this point). We have added a comment on the potential of MD simulations to get more information about the mechanism of Phe397 displacement in lines 389-393:

- *Theoretical Study of the Mechanism of the Hydride Transfer between Ferredoxin-NADP⁺ Reductase and NADP⁺: The Role of Tyr303. Journal of the American Chemical Society 2012 134 (50), 20544-20553. DOI: 10.1021/ja310331v*

- *Peregrina, J.R., Lans, I. & Medina, M. The transient catalytically competent coenzyme allocation into the active site of Anabaena ferredoxin NADP⁺-reductase. Eur Biophys J 41, 117-128 (2012). <https://doi.org/10.1007/s00249-011-0704-5>*

3. Comparing the structure of wt spNOX and F397A mutant, there is little structural change of spNOX protein, except the conformation of the nicotinamide group of NADPH substrate. Would this reflect what happens during the catalysis of wt spNOX?

Currently, there isn't any structure available of a WT FNR family member in a productive conformation. However, the reaction mechanism of *Anabaena* ferredoxin-NADP⁺ reductase has been extensively studied, with a collection of WT and mutant enzymes complexed with NADP⁺ available, and several theoretical studies with molecular dynamic simulations (see above) based on these structures. These simulations in *Anabaena* FNR reveal that major changes occur in the distances and orientations between the rings of Tyr303, NADP⁺ and FAD. However, other relevant residues like Ser80, Cys261 and Glu301 only show minor rearrangements that might contribute to shorten the distance between the coenzymes.

Contrary to FNR Tyr303Ser mutant, which has a higher affinity for NADP⁺ than the WT and can only catalyze hydride transfer from NADPH to FAD, our SpNOX Phe397Ala mutant (and also the Phe397Ser mutant described in Petit-Härtlein et al. BioRxiv) does not show any increase in the

apparent affinity for NADPH and is able to catalyze hydride transfer faster than the WT. Besides, apart from the major changes in the position of the nicotinamide ring of NADPH, SpNOX Phe397Ala shows a rearrangement in the sidechain of Glu395 (equivalent to FNR Glu301). Compared to the inactive NADPH-bound WT structure, Glu395 interacts with nicotinamide N7N via an H-bond, probably contributing to the shortening of the distance between the nicotinamide and the isoalloxazine rings.

Therefore, considering that Phe397Ala SpNOX is able to catalyze hydride transfer, and that MD simulations of WT *Anabaena* FNR only show major changes in the position of the isoalloxazine, nicotinamide and C-terminal hydroxyphenyl rings, but not any other major rearrangements, the structure of Phe397Ala SpNOX may well reflect what happens during catalysis in WT SpNOX. Nevertheless, as discussed above, an in-depth study combining MD simulations with time-resolved cryo-EM studies could help to get further details, but as we also indicated above, in our belief it is out of the scope of the manuscript.

4. The sequence alignment around F397 in sFig. 8 is misleading to some extent. From Sfig.8, it seems that F397 is at the equivalent position to F1544 in DUOX1. But F1544 is on a stable beta structure in DUOX1. F397 is more like the F1551 at the C-terminus of DUOX1. I would suggest using more advanced sequence alignment tools, such as PROMALS3D for this tricky alignment.

We thank the reviewer for the suggestion. We have repeated in the revised version of the manuscript all alignments with PROMALS3D (see revised Supplementary Figs. 8a-c, and Supplementary Fig. 11). Of note, the alignment to DUOX1 is now more consistent with the structural picture.

5. In line 270, the authors conclude aromatic residues are not necessary for electron transfer between two hemes because the F107L mutation did not affect the activity of spNOX. Is it possible that in the F107L mutant, the conformation of Y136 at the same layer is somehow changed to mediate electron relay between two hemes? The authors might want to check F107 and Y136 double mutant.

Electron transfer rates between redox centers can be calculated by the quantum version of Marcus electron transfer theory, which depends on the edge-to-edge distance between the redox centers, the nuclear reorganization energy upon electron transfer (λ), the driving force for electron transfer (redox potentials) and the atom-packaging density (ρ) of the protein medium (see Page C. et al. Nature. 1999). On the basis of the multiple structural studies of redox proteins available, it has been found that redox proteins usually show designs that maintain redox centers at distances of ~ 14 Å or less, with most distances being between 6 and 10 Å. This suggests that redox proteins have evolved to have a structure in which redox centers are separated by distances below 14 Å, without having to optimize other factors that may influence electron-transfer rates. When these distances are kept, electron transfer by tunneling reaches rates fast enough not to be a rate-limiting step for the overall reaction (see Page CC. et al. Curr. Opin. Chem. Biol. 2003 below).

For eukaryotic NADPH oxidases and SpNOX, distances between the cofactors mediating single-electron-transfer steps (i.e. FAD and hemes) are maintained in the 8-11 Å range.

These distances should be short enough for electron transfer not to be rate-limiting, independent of the protein sequence. The distance between hemes and FAD-inner heme in SpNOX and other NADPH oxidases is between 8 and 10 Å and meets this criterion. For hemes of similar redox potentials separated by a distance of 10 Å, and assuming values of λ and ρ consistent with the literature (Page et al. 1999), Marcus's theory predicts an electron transfer rate higher than 10^6 s⁻¹; many orders of magnitude larger than the measured turnover rates of SpNOX or any related catalyst. In this context, it is hard to imagine that conserved residues meaningfully affect SpNOX activity via an effect on inter-heme electron transfer rates.

This is in line with our result for the F107L mutant, for which we did not observe any important change in the catalytic properties. Besides, as suggested by this reviewer, a double mutant with F107L and Y136L has been generated and its steady-state kinetics calculated. The mutant shows a similar activity to WT and F107L, suggesting that neither F107 nor Y136 residues are crucial in the electron transfer between the two heme cofactors.

In this context, to address the concern of the reviewer, the Michaelis-Menten-fitted curve of the double mutant has been added in the revised version of Supplementary Fig. 14 (panel 14f). Therefore, we think that the aromatic residue conserved in NADPH oxidases is not an essential path for efficient electron transfer, and a reduction of protein activity in eukaryotic NADPH oxidases when this residue is mutated could reflect structural changes. Additionally, these results have been discussed in lines 286-297 of the revised version of the manuscript.

Page, C., Moser, C., Chen, X. et al. Natural engineering principles of electron tunnelling in biological oxidation–reduction. Nature 402, 47–52 (1999). <https://doi.org/10.1038/46972>

Page CC, Moser CC, Dutton PL. Mechanism for electron transfer within and between proteins. Curr Opin Chem Biol. 2003 Oct;7(5):551-6. doi: 10.1016/j.cbpa.2003.08.005. PMID: 14580557.

6. Between FAD and inner heme, there are a few aromatic residues? Do they participate in the electron transfer between FAD and inner heme?

As stated in the previous answer, there is no need for SpNOX to have specific residues mediating an efficient electron transfer between FAD and the inner heme. Besides, these residues are part of the FAD-binding lobe (e.g. Tyr122), or seem to be mediating the docking site between the dehydrogenase (DH) domain and the transmembrane (TM) domain (e.g. Phe399). Therefore, we expect that mutations in these residues lead to alterations in FAD binding (FAD structure, FAD-binding affinity or even FAD-redox potential), and/or in DH and TM-domain docking. This would affect parameters related to electron tunneling rates like edge-to-edge distances, driving force, nuclear reorganization or packing density, but would not mean that these residues are a preferred nor essential path for these electrons.

In this context, to address the concern of the reviewer, we have produced a C-terminal-deletion SpNOX mutant (Δ 398-400) and analyzed its steady-state kinetics with NADPH as substrate (see Supplementary Fig. 14a). We observe a modest increase in K_M and a modest decrease in k_{cat} . This leads to a reduction of the catalytic efficiency to half with respect to the WT. As both parameters seem to be affected, we think that the removal of the tail may be affecting the docking of the DH and TM domains, and NADPH-binding as a consequence. These results have been included in the revised version of the manuscript (lines 236-240) and as a new panel in Fig 2 (panel e). We have not generated mutants of Tyr122 or Trp125 because we think they are involved in FAD-binding and most probably in the folding of the TM domain too, so the results (likely a reduction in apparent k_{cat}) would not be conclusive. In fact, a Tyr122Ala mutant has been characterized in the SpNOX study of Petit-Härtlein et al, and shows a six-fold reduction in the affinity for FAD and a reduction in k_{cat} . Thus, a comment regarding this mutant has been additionally included in the revised version of the manuscript (lines 226-228). Besides, an Arg126Ala mutant was characterized in the Petit-Härtlein et al study. This mutant shows a strongly decreased affinity for FAD, in spite of interacting directly only with the inner heme in our high-resolution structures. This suggests that the amino acids involved in FAD and inner-heme binding are relevant to preserve the whole DH and TM domain interface, and therefore mutations in these amino acids would compromise the structure and catalytic activity of SpNOX.

Petit-Härtlein et al. X-Ray Structure and enzymatic study of a Bacterial NADPH oxidase highlight the activation mechanism of eukaryotic NOX. BioRxiv. 2023.

Minor issues:

1. The outer heme of spNOX is in a different pose compared to other NOX members. The author might elaborate more on this.

We thank the reviewer for this comment. To address the concern of the reviewer, we have updated Supplementary Fig. 9 with a comparison of the outer hemes. Additionally, we added a comment about this difference in the Results section, where the TM domain of SpNOX is compared to the one of eukaryotic NOXs (see lines 202-205). Besides, we have highlighted the residues mediating the interaction with the propionate groups of the outer heme in the revised version of Supplementary Fig. 8a.

2. CBC atom of outer heme (507) has an alternative fitting which might be better than the current one.

We thank the reviewer for this comment. We have checked and updated the models according to the suggestion of the reviewer.

Reviewer #2:

Remarks to the Author:

Dubach, Acosta, and Murphy present a comprehensive analysis of the cryoEM structure of a bacterial NADPH oxidase (NOX), shedding light on its mechanism. NOXs are membrane-bound enzymes producing reactive oxygen species (ROS) and utilize intracellular NAD(P)H to reduce oxygen, generating superoxide or H₂O₂ outside the cell. The authors exhibit exemplary experimental prowess, achieving high-resolution maps for a protein of less than 50 kDa—an experimental feat that underscores their expertise.

From a biochemical perspective, a pivotal finding emerges regarding NADPH binding. Based on existing literature, the authors engineer a mutant, facilitating the elucidation of the enzyme structure bound to nicotinamide in proximity to flavin. Consequently, the study establishes that NOXs operate through hydride transfer, debunking previous suggestions and aligning with the extensive knowledge based on ferredoxin reductases.

The structural analysis confirms the NOX structural conservation in dehydrogenase and trans-membrane domains, along with well-preserved FAD- and heme-binding sites. Notably, the oxygen-reacting center exhibits divergence from counterparts in other bacterial and human NOXs. The authors speculate that their protein may not be a conventional NOX but could function in the reduction of another protein. It could potentially serve as a ferric ion reductase—a notion left unexplored in the text and warrants experimental validation. Can the authors experimentally verify the protein activity as an iron reductase? Does the genomic context suggest the function?

In essence, this work contributes to our understanding of NOXs, particularly by clarifying the flavin reduction mechanism and affirming known structural features. However, the true substrate of the enzyme remains elusive. While endorsing the publication in journals such as Nature Communications, the manuscript may lack the depth and groundbreaking innovation typically expected for a Nature Structural Biology publication.

We thank the reviewer for his/her work, and these useful comments that have been addressed in the revised version of the manuscript.

In the original study describing SpNOX, ferric reductase activity was examined and found to be comparable to the ferric reductase activity of human NOX2 (Hajjar *et al.* 2017). This assay was performed under aerobic conditions and in the presence of superoxide dismutase (SOD) to minimize the reduction of Fe(III) by the produced superoxide. The amount of Fe(III) reduction with SOD present was minimal, and the idea of SpNOX being a true ferric reductase was discarded. However, in this paper we show that SpNOX can directly transfer electrons to SOD, invalidating the previous activity assay results carried out in its presence, as SOD would also act as a competitor to the ferrous iron at high concentrations, in addition to scavenging any superoxide formed. Moreover, in the revised version of the manuscript, we show SpNOX does exhibit direct electron transfer to ferrous iron by performing the assay under anaerobic conditions (added as Supplementary Fig. 1h). However, the reduction rate (3.6 min⁻¹) is 40-times slower than for cytochrome c (124.7 min⁻¹) or 10-times slower than atmospheric oxygen (48 min⁻¹, see lines 163-164). We further performed Michaelis-Menten kinetics analysis for the Fe(III) reduction (added as Supplementary Fig 1i), showing a low apparent k_{cat} (0.05 s⁻¹) and a high apparent K_M (81.3 μM) which result in a very low catalytic efficiency (k_{cat}/K_M , 0.00063 s⁻¹ μM⁻¹) (see lines 165-169). Altogether, these results speak strongly against Fe(III) as the physiological electron acceptor.

We have carried out an analysis of the genomic context of the SpNOX gene (SPNHU17_RS03015, see attached image) in one of the available *S. pneumoniae* reference genomes at NCBI (NZ_CP020549.1). The SpNOX gene is located near a vancomycin resistance operon (*vncR*, *vncS*), the glycolytic fructose bis-phosphate aldolase gene (SPNHU17_RS03010) and an operon

of ABC transporters and permeases potentially involved in glutamine transport (SPNHU17_RS03020, RS03025 and others). Although we cannot rule out a meaningful connection to the function of any of these genes, a general connection to known or putative reactivity of SpNOX is not apparent.

The position of SpNOX gene SPNHU17_RS03015 in the *S. pneumoniae* genome

Hajjar, C., et al., *The NOX Family of Proteins Is Also Present in Bacteria*. mBio, 2017. **8**(6): p. e01487-17.

A link to the genomic location of SpNOX:

https://www.ncbi.nlm.nih.gov/gene?cmd=retrieve&list_uids=66805763

Reviewer #3:

Remarks to the Author:

A. Summary of the key results

We thank the reviewer for his/her work, the realization of the interest of the work, and his/her suggestions to improve the presented work. Moreover, as suggested, we have updated the Abstract to give more specific information about the findings of our study and their relevance to the redox community.

B. Originality and significance: if not novel, please include reference

There is a similar PsNOX study published in

BioRxiv: <https://www.biorxiv.org/content/10.1101/2023.10.16.562591v1>

C. Data & methodology: validity of approach, quality of data, quality of presentation

Good, except for the kinetic data.

D. Appropriate use of statistics and treatment of uncertainties

Kinetic data should use SD iso SEM.

We have made these changes, thank you.

E. Conclusions: robustness, validity, reliability

See my text.

F. Suggested improvements: experiments, data for possible revision

See my text.

G. References: appropriate credit to previous work?

OK except for not mentioning the study of Petit-Härtlein et al.

We thank the reviewer for this point. The study of Petit-Härtlein et al. was uploaded to BioRxiv after we submitted the manuscript to NSMB, so that it was not possible for us to include a comparison to this study in our original manuscript. We have now referred to this study in the following parts of our revised version of the manuscript, as we think it adds relevance to our results:

- The existence of a manuscript describing a high-resolution structure of the DH domain of SpNOX and a low-resolution structure of full-length SpNOX Phe397Trp is mentioned in lines 182-185 of the revised manuscript.
- The high-affinity of SpNOX for FAD, and its lower affinity for smaller, less bulky flavins like flavin mononucleotide (FMN), is mentioned in lines 217-218, where we describe the FAD-binding site.
- The Tyr122Ala mutant characterized in that study is referenced in lines 226-228 to confirm the relevance of Tyr122 in FAD binding.
- The discussion made in that study regarding the activation of eukaryotic NOXs to achieve hydride transfer by tensioning of the NADPH-binding lobe is mentioned in lines 274-282, and supported by the new panels 13d-f of Supplementary Figure 13 comparing NADPH-binding in SpNOX and high-calcium human DUOX1.

- In the Petit-Härtlein et al. study, two Phe397 mutants were characterized: a Phe397Ser mutant with equivalent steady-state kinetics to our Phe397Ala mutant; and an inactive Phe397Trp mutant. We have mentioned these mutants in the Results (lines 182-185 and 323-325) and Discussion (lines 389-392) sections, where we describe the potential hydride transfer mechanism of SpNOX.

It is important to note that the study by Petit-Härtlein et al. did not achieve comparable resolution values to ours. This fact has been a limiting factor and has led to incorrect interpretations about the role of some residues, including Arg126 and Tyr136. Other differences, like a further distance between Phe397 and the isoalloxazine ring in their high-resolution structure of the DH domain, could be explained as a consequence of the missing TM domain. This domain is involved in FAD binding by Tyr122, and in positioning the C-terminal tail orthogonally to the membrane. Additionally, the adenine of the FAD has been misplaced in the full-length structure due to taking the FAD geometry from the high-resolution DH domain structure, which lacks the crucial residues for correct binding (i.e. Tyr122). Due to the fact that the work of Petit-Härtlein et al. does not include any full-length nicotinamide-bound structure, so that a detailed comparison is not possible.

H. Clarity and context: lucidity of abstract/summary, appropriateness of abstract, introduction and conclusions

See more details here below.

Dubash et al.'s manuscript marks a significant milestone in unraveling the substrate-bound structures of SpNOX, providing crucial insights into its catalytic mechanism. The study underscores a pivotal transient displacement of a C-terminal aromatic residue, a key element for hydride transfer—a phenomenon shared among members of the FNR superfamily. Noteworthy is the observation that the SpNOX Phe397Ala mutant adopts a productive conformation, correlating with enhanced turnover rates. Intriguingly, a parallel catalytic role has been suggested for a strictly conserved Phe residue at the C-terminus of eukaryotic NOXs, yet so far no published structures have illustrated its interaction with FAD.

Technically, the manuscript showcases the determination of high-resolution cryo-EM structures for the relatively compact SpNOX (46 kDa), underscoring advancements in microscopy, detectors, and software that propel the boundaries of single-particle cryo-EM.

While the narrative is compelling, the manuscript falls short in delving into the intricate details of substrate-bound structures and lacks a robust comparison with other NOXs in the discussion. Addressing this gap in the next review round will undoubtedly enhance the manuscript's depth and impact.

We thank the reviewer for this comment. We have revised the manuscript according to the suggestions of the reviewer, and now the following figures and parts of the text compare SpNOX with other NOXs, giving details about specific structural aspects:

→In the "*Architecture of SpNOX*" section of the *Results*:

- A general comparison of the global domain arrangement and sequences of SpNOX and the other NOXs is described in lines 189-193, and supported by Fig. 1 and Supplementary Fig. 8a-c.
- A comparison between the binding to the outer heme between SpNOX, eukaryotic NOXs and csNOX5 has been added to lines 202-205, and Supplementary Figs. 8a and 9.
- A detailed comparison between the whole DH domains and the FAD-binding regions is made in Supplementary Fig. 8a-c, 9, 10d-f and in lines 206-216 and 217-233.

→ In the section “*The NAD(P)H-binding site and the electron transfer pathway of SpNOX*” of the *Results*:

- A detailed description of the NAD(P)H-binding site of SpNOX is given in lines 244-253 and lines 254-282.
- To our knowledge, the only published study of a NOX in a substrate-bound configuration is for human DUOX1. There are also structures of other members of the FNR superfamily with NADPH bound. In lines 259-270, we compare the NAD(P)H-binding site of SpNOX to the sites of high-calcium human DUOX1, *Anabaena* FNR, and the predicted site of NOX2, to explain in detail the lack of NADPH specificity in SpNOX. These descriptions and comparisons are supported by Figures 3a-b, Supplementary Fig. 8a-c and Supplementary Fig. 9.
- We have now added a more detailed comparison between the NADPH-binding sites of human DUOX1 and SpNOX, and added a reference to the Petit-Härtlein et al. preprint (lines 274-282 and Supplementary Fig. 13, including new panels 13d-f).
- A comparison between the hydrophobic cluster present between hemes in SpNOX and the other NOXs, is made in lines 286-300.
- The search of potential oxygen-binding sites is described in lines 334-352.

→ In the Discussion, we made comparisons between SpNOX and eukaryotic NOXs focused on the role of the C-terminal aromatic residue (lines 384-404), and on the potential oxygen reduction centers (lines 410-422). These discussions are supported by Figs. 3 and 4, and Supplementary Figs. 9 and 10.

Several specific points merit attention. While the manuscript is well-crafted, it neglects a recent SpNOX study by Petit-Härtlein et al., which utilized X-ray crystallography and was uploaded to BioRxiv 3 days before the work of Dubash et al. A comparative discussion of these akin results, despite not sharing the exact same interpretation, could substantially enhance the manuscript's relevance to the NOX and redox communities. I recommend facilitating communication between both research groups to foster collaboration and potentially refine interpretations.

Please see our response above.

Furthermore, clarifications are needed on the inaccurate claim of revealing new details on catalytic activity and electron-transfer pathways, as these have been previously demonstrated in eukaryotic NOXs.

In the text, we already mention that the proposed electron-transfer pathway in SpNOX is similar to the previously described in other NOXs (lines 283-285). However, we do think that we have also made several new contributions that add relevant information not only for the electron-transfer pathway, but also for the catalytic mechanism. Here, we highlight the following specific points, addressed in the manuscript, which inform our understanding of catalytic activity and electron transfer within NADPH oxidases:

- Structural and mutational evidence for the role of Phe397 in the reaction. These experiments confirm that NOX enzymes use a hydride transfer mechanism for electron transfer from NAD(P)H to FAD, like other members of the FNR superfamily (lines 301-333).
- Analysis of the potential role of the C-terminal tail residues in the docking of the DH and TM domains (added to the revised version of the manuscript, lines 236-240).
- Mutational analysis of the (absence of) role of Phe107 or Tyr136 in efficient electron tunneling between heme groups, in agreement with the quantum version of Marcus electron transfer theory (lines 292-300).

-Structural and mutational analysis of substrate specificity, which gives residue-level insights into the mechanism of NADPH and NADH binding in SpNOX (lines 259-270). Three more technical replicates (same protein batch) were added to the kinetic characterization of the Y353R SpNOX mutant to get a better fitted curve and more precise kinetic parameters (Supplementary Fig. 14c).

Including a schematic overview of the hydride transfer mechanism

To address the concern of the reviewer, we have included that schematic overview as Supplementary Fig. 15 and mentioned it in the Discussion section of the revised version of the manuscript (line 392-393).

and explaining abbreviations like DH in the main text are essential for reader comprehension.

We have carefully checked that we define all abbreviations when they are first used.

Graphical and data representation concerns include the need to adjust Fig. 1b's x-axis to 50 μ M for better visibility of the fitting in the crucial area. Fig. 2e requires improved clarity in the legend, necessitating adjustments in terms of a title that conveys a clear message.

To address this issue, we have adjusted the x-axis in Fig. 1b with a gap between the region of 100 and 250 μ M for a better visibility of the fitting at lower substrate concentrations.

The steady-state kinetics data presented in Figure S14 necessitate accurate fitting using Hill coefficient equations.

Given that spNOX is a monomeric protein, and given that our structural analysis in the presence of NADPH revealed a single NADPH-binding site, it is difficult to imagine that there are any relevant cooperativity effects. For this reason, we think a Hill coefficient of 1 is appropriate in this context.

Additionally, there is an issue with the correctness of the KM units that requires attention and correction.

Fixed in the revised version of the manuscript. Thank you.

Crucial details on obtaining the AlphaFold model are missing from the methods section,

The AlphaFold2 models shown in this work were downloaded from the AlphaFold Protein Structure Database created by DeepMind in collaboration with the EMBL.

To address this issue, we have cited this paper, where specific details for the generation of the models can be found, in the methods section (lines 643-644).

Varadi M., et al. AlphaFold Protein Structure Database: massively expanding the structural coverage of protein-sequence space with high-accuracy models, Nucleic Acids Research, Volume 50, Issue D1, 7 January 2022, Pages D439–D444

and mechanistic intricacies, especially regarding electron transfer and FAD accepting two hydrides while NAD(P) accepts only one, remain unclear and merit detailed discussion.

We are not aware of precedent for the idea that FAD can accept two hydrides, and a careful analysis of the scientific literature did not reveal any studies suggesting that it can. Perhaps the reviewer meant to say 'FAD accepts two electrons'?, in which case we agree; in particular, the

precedent established for redox reactions involving nicotinamide as substrate is that a hydride (two electrons and one proton) is transferred to the flavin, which can undergo two-electron (hydride) and one-electron transitions.

In this context, to address this issue, we have added in the revised version of the manuscript a scheme of the proposed catalytic cycle of SpNOX in Supplementary Fig. 15.

Finally, a comprehensive exploration of the oxygen reduction mechanism, including speculation on potential two oxygen binding sites and electron tunneling, is warranted for a more thorough understanding of the manuscript's implications. Addressing these points will undoubtedly elevate the manuscript's overall quality and impact of this interesting study which I would like to see published in this journal.

We thank the reviewer for this comment. The two potential oxygen-binding sites, how they were analyzed, and the mutagenic analyses are included in the Results section of the manuscript (lines 334-352) and in the Discussion section (412-437). Although the results of these analyses do not allow us to draw confident conclusions on the nature or binding site of the final acceptor, we think that they point towards a role for SpNOX different from ROS production and more related to the reduction of other substrates, most probably bulky substrates like proteins. Regarding electron tunneling from FAD to the transmembrane hemes, we have produced a new double mutant (F107L/Y136L) for the revised version of the manuscript (lines 290-300) and new Supplementary Fig. 14f), and we have concluded that these aromatic residues between hemes are not a preferred route for electron tunneling in SpNOX. This is agreement with previous studies (see *Page C. et al* below) showing that distances below 14 Å between redox centers transferring single electrons are enough to reach electron-transfer rates (calculated by the quantum version of the Marcus theory of electron tunneling) several orders of magnitude faster than redox reactions.

Page, C., Moser, C., Chen, X. et al. Natural engineering principles of electron tunnelling in biological oxidation–reduction. Nature 402, 47–52 (1999). <https://doi.org/10.1038/46972>

Page CC, Moser CC, Dutton PL. Mechanism for electron transfer within and between proteins. Curr Opin Chem Biol. 2003 Oct;7(5):551-6. doi: 10.1016/j.cbpa.2003.08.005. PMID: 14580557.

Final Decision Letter:**Message:** 6th Jun 2024

Dear Dr. Murphy,

We are now happy to accept your revised paper "Structural and mechanistic insights into *Streptococcus pneumoniae* NADPH oxidase" for publication as an Article in Nature Structural & Molecular Biology.

Your paper will be published online soon after we receive proof corrections and will appear in print in the next available issue. You can find out your date of online publication by

contacting the production team shortly after sending your proof corrections.

Please note that *Nature Structural & Molecular Biology* is a Transformative Journal (TJ). Authors may publish their research with us through the traditional subscription access route or make their paper immediately open access through payment of an article-processing charge (APC). Authors will not be required to make a final decision about access to their article until it has been accepted. Find out more about Transformative Journals

Sincerely,
Sara

Sara Osman, Ph.D.
Senior Editor
Nature Structural & Molecular Biology